# DNA T-shaped crossover tiles for 2D tessellation and nanoring reconfiguration

Qi Yang[1], Xu Chang[1], Jung Yeon Lee[1], Minu Saji[1] & Fei Zhang [1] ✉

DNA tiles serve as the fundamental building blocks for DNA self-assembled nanostructures such as DNA arrays, origami, and designer crystals. Introducing additional binding arms to DNA crossover tiles holds the promise of unlocking diverse nano-assemblies and potential applications. Here, we present one-, two-, and three-layer T-shaped crossover tiles, by integrating T junction with antiparallel crossover tiles. These tiles carry over the orthogonal binding directions from T junction and retain the rigidity from antiparallel crossover tiles, enabling the assembly of various 2D tessellations. To demonstrate the versatility of the design rules, we create 2-state reconfigurable nanorings from both single-stranded tiles and single-unit assemblies. Moreover, four sets of 4-state reconfiguration systems are constructed, showing effective transformations between ladders and/or rings with pore sizes spanning ~20 nm to ~168 nm. These DNA tiles enrich the design tools in nucleic acid nanotechnology, offering exciting opportunities for the creation of artificial dynamic DNA nanopores.

DNA has been demonstrated to be a reliable building block for bottom-up assembly of nanostructures with high programmability and accuracy[1,2]. The developments of structural DNA nanotechnology have advanced fundamental studies for technological applications far beyond its biological function of carrying genetic information. For example, several challenging scientific topics have been explored with the help of DNA nanotechnology, such as the mechanical studies of single biomolecules[3], artificial enzyme cascades[4,5], dynamic DNA nanopores[6,7], lipid remodeling[8,9], and synthetic nanostructures for light harvesting[10,11]. Therefore, engineering DNA self-assemblies with programmable behaviors is essential for the development of DNA nanotechnology and its diverse applications.

To assemble DNA nanostructures from simple modules, DNA tile-based approaches are well suited since the repeating tile units and the interactions between tiles can be designed and programmed systematically[12]. Guided by sticky-end pairing rules, monomer DNA tiles link together and grow into one-dimensional (1D), 2D, and 3D structures. The antiparallel double-crossover (AX) tile is one of the classic designs as well as the most widely adopted class of tile that has been used for DNA assembly[13]. To introduce additional interaction directions of AX tiles, different designs of DNA tiles have been studied. The multi-arm branched motifs utilized rigid AX tiles as arms and

controlled the angle between adjacent arms with polyT loops[14–16]. Another interesting way to create multi-directional interactions was using T-junction tiles[17,18], which contained two perpendicular arms and a loop-strand sticky end. However, classic T-junction tiles can only assemble into 2D arrays on surfaces due to the flexibility of the tiles. All these studies suggest that it is critical for a DNA tile to have programmable directional interactions as well as sufficient rigidity, enabling solution-based self-assembly of large target nanostructures.

In this work, we create a category of DNA tiles, named T-shaped crossover (TC) tiles, that integrate the structural features of T-shape junctions with AX tiles. The TC tiles retain the rigidity of AX tiles and contain an additional arm from T-junction infusion. Fourteen different one-layer TC tiles are designed and studied to investigate the factors that affect linear or 2D array formation. Then we present additional six-layered TC designs, manifesting their capabilities to form ladders, tubes, and lattices. To illustrate the adaptability of the design rules, we use the single-stranded tile (SST) approach to construct a pair of reconfigurable nanorings with predetermined dimensions and addressable surfaces. Further, four pairs of 2-state nanorings are constructed from TC tiles with tunable pore sizes ranging from ~36 nm to ~55 nm, ~54 nm to ~82 nm, ~41 nm to ~62 nm, and ~61 nm to ~94 nm.

[1]Department of Chemistry, Rutgers University, Newark, NJ 07102, USA. ✉ e-mail: fei.zhang@rutgers.edu

Finally, we demonstrate four 4-state reconfigurable systems that adopt different configurations between ladders and/or rings with pore sizes spanning ~20 nm to ~168 nm. In addition, we also study the intermediate states in the different reconfiguration processes.

## Results and discussion

### Design of TC tiles

Inspired by the T-junction tiles with an intrinsic perpendicular arm[17], we proposed the TC tiles by infusing T junctions into antiparallel double-crossover tiles to form different angles and orientations. As shown in Fig. 1a, one-layer TC tiles adopted a T-junction loop (T-loop) into their major groove between the crossovers vertically or horizontally, so that they inherited the perpendicular arms while retaining the rigidity of double helical structures. The T junction comprising a 6-nucleotide loop also worked as a sticky end that hybridized with a complementary single strand through Watson–Crick base pairing. The combination of two pairs of a T-loop-sticky end and one pair of a double T junction's T-loop with a sticky end allowed us to program distinct target patterns by changing sticky-end matching rules. The perpendicular geometry and T-loop sticky-end interaction made the TC tile a versatile construction module. Similarly, two- and three-layer TC tiles were created by rotating the double helices and inserting the perpendicular arm between crossovers, respectively.

Moreover, a C-shape TC tile (C-tile, Fig. 1b) was employed to create finite-sized nanoring structures. Based on the geometry of target nanorings, the horizontal arms of the C-tile monomers were designed with inequivalent lengths. Unlike the previously published DNA nanorings with a single helical structure[19–22], the C-tile-based nanorings had rigidified multiple helices that allow modifications on nanorings with functional entities. As an example, we demonstrated geometrical reconfiguration of assembled nanorings by anchoring toehold-mediated strand displacement reaction (SDR) functional domains to the horizontal arms[23–25].

### One-layer TC tiles

We began the study of TC system by designing fourteen single-layered tiles (all the DNA helices were in the same plane, as single-layered, Fig. 2a–h and Supplementary Figs. 1–19) to form ladders or grids through programable sticky-end matching rules. The connection between TC tiles involved loop-strand interaction (Bn to Bn*, Fig. 2) and hybridization between complementary strands (Sn to Sn*, Fig. 2). Self-complementary sticky ends were employed to minimize the intrinsic curvatures and possible mismatches between individual tiles[26]. As shown in Fig. 2a, TC-1–1, a cross-shaped tile, was achieved by fusing one end of AX arms with a T junction. The sticky-end matching rules was programed to include two horizontal bindings (S1 to S1*, B1 to B1*) and one vertical self-complementary sticky end (Sn) to enable ladder formation. Gel images indicated the formation of large nanostructures (Supplementary Fig. 1c). Atomic force microscopy (AFM) images revealed the ladder conformation (Fig. 2a and Supplementary Fig. 1d). By modifying the sticky-end matching rules and changing arm lengths, a tile TC-1-2 was created for 2D assembly (Fig. 2b and Supplementary Fig. 2). Interestingly, the matching rules assigned in TC-1-2 enabled two types of potential interactions between the T-loop and the sticky end (Supplementary Fig. 3). One interaction was the binding between two monomers (intermolecular binding) to form array patterns as designed. The other interaction was the possible self-binding within one monomer (intramolecular binding) to form a rectangular tetramer because the T-loop and the sticky end were close to each other in space (Supplementary Fig. 3a). The native gel results of TC-1-2 showed a large band on top, and no tetramer band was observed. AFM images revealed micrometer-sized unwrapped/wrapped tubular arrays, indicating that the T-loop and sticky-end pair preferred intermolecular binding under our experimental conditions.

To explore the factors that affect the linear and 2D array formation of one-layer TC tiles, we explored different structural designs in the TC system, including struts between perpendicular arms, double

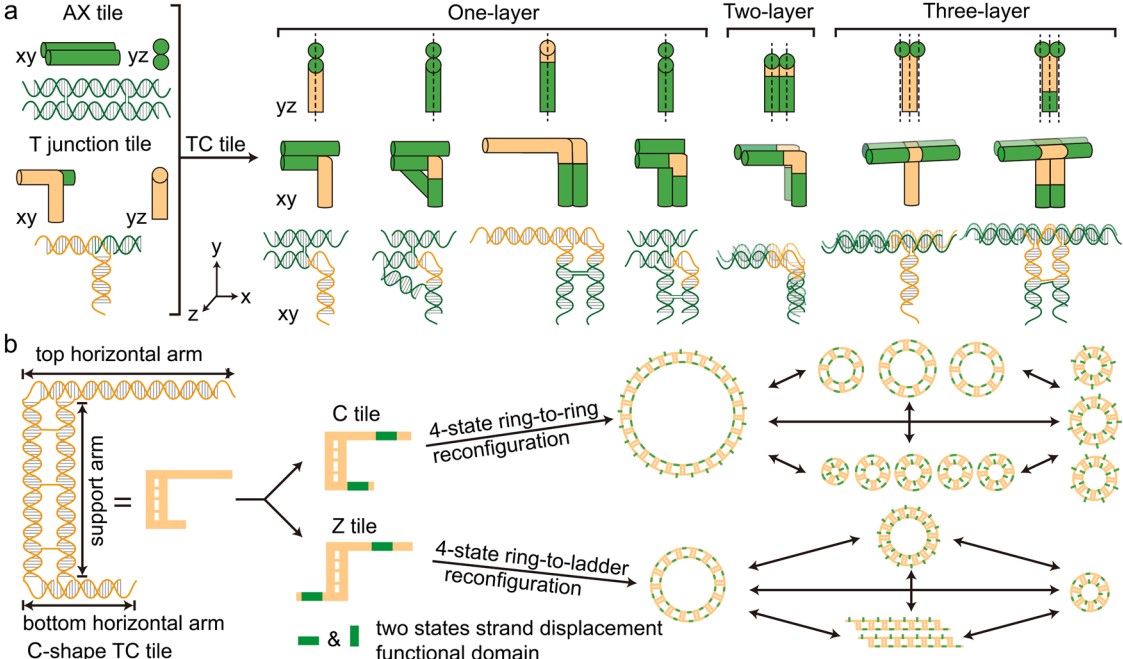

**Fig. 1 | Schematics of DNA TC tiles and reconfigurable DNA nanorings. a** The design of one-, two-, and three-layer TC tiles. These TC tiles adopt AX crossover and T-junction features. Crossovers are placed between two helices to reinforce the arms of TC tiles (green). T junction introduces a third direction of monomer tile to create a perpendicular shape (orange). The dash lines in side view indicate the layers of tiles. **b** Two types of the reconfigurable 4-state DNA nanostructures. Both C- and Z-shape tiles contain toeholds on horizontal arms. The toehold-mediated strand displacement reaction (SDR) enables the shortening or lengthening of the horizontal arms. Assembled structures from Z or C tiles can reconfigure to adopt four different geometric states. For example, the C-tile is employed for 4-state ring-to-ring reconfigurations, and the Z-tile is for 4-state ladder-to-ring transformations.

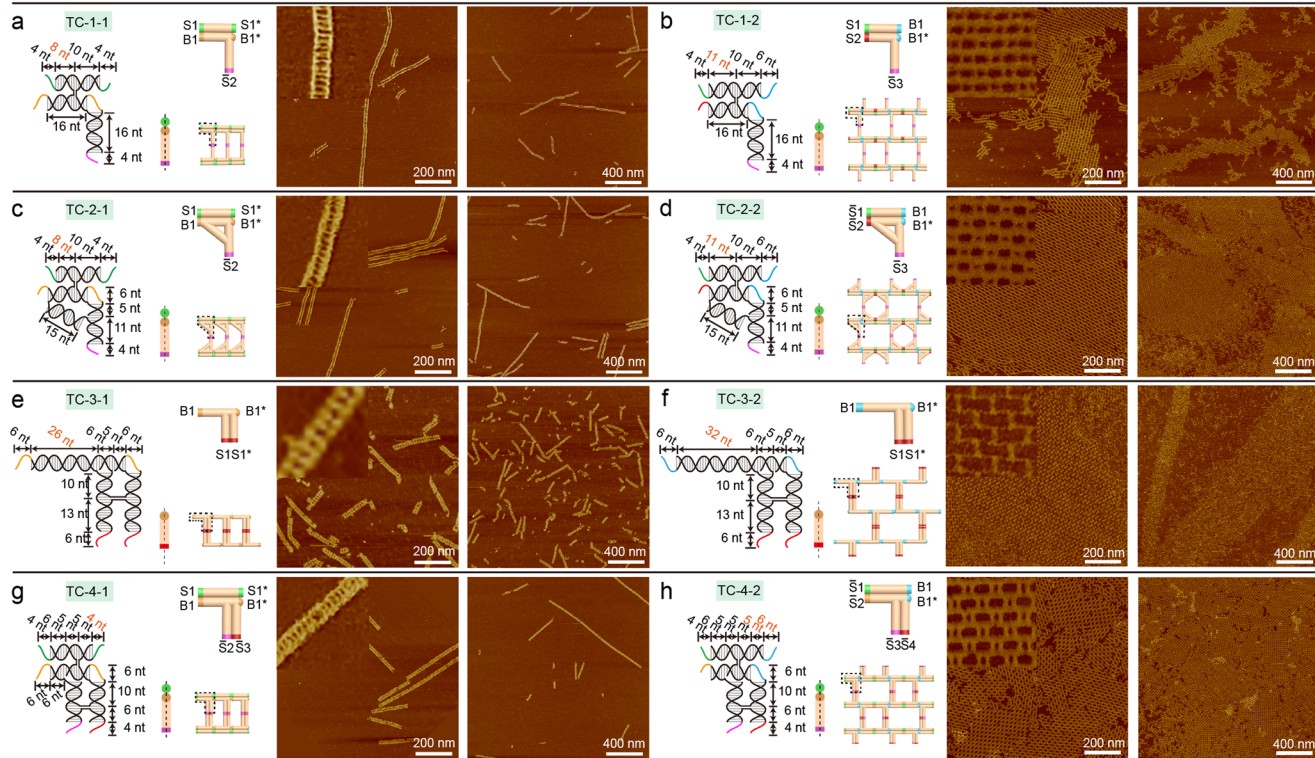

**Fig. 2 | Design schematics, sticky-end matching rules, and corresponding AFM images for one-layer TC tiles. a** TC-1-1; **b** TC-1-2; **c** TC-2-1; **d** TC-2-2; **e** TC-3-1; **f** TC-3-2; **g** TC-4-1; **h** TC-4-2. Each figure contains the design of tile, cylindrical model of tile (side view and 3D view) and AFM images of assembled arrays. The dash lines in a side view indicate the layers of tiles. For the sticky-end matching rules, a pair of complementary sticky ends are labeled as Sn and Sn* (*n* is a number), and a T-loop sticky-end interaction is marked as Bn (sticky end) and Bn* (T-loop). Each pair of complementary sticky ends or interaction between a T-loop and a sticky end is given a unique color identifier. A self-complementary sticky end is denoted as Sn. All the tiles labeled as Tile n-1 target ladders, while Tile n-2 target 2D grids. All the AFM images are taken after 66 h annealing program. Inset image size: 100 nm × 100 nm. Source data are provided as a Source Data file.

T-junction design (Supplementary Fig. 4), crossovers on arms, assembly units, and sticky-end length. TC-2-1 and TC-2-2 were created by introducing a short, double helical strut between two adjacent arms. Notably, unlike the reported multi-arm tiles with struts[27], no spacer nucleotides were inserted into the joint position between struts and arms (Fig. 2b and Supplementary Fig. 2), thus creating a compact and rigid monomer. In Fig. 2c and Supplementary Fig. 5, we observed ladders that were similar to TC-1-1. The micrometer-sized 2D arrays yielded from TC-2-2 were generally larger than those from TC-1-2, suggesting that the struts helped to define the angle between arms and also corrected possible curvature of the structures[27].

We next studied the double T-junction designs by creating TC-3-1 and TC-3-2 tiles (Fig. 2e, f and Supplementary Figs. 7 and 8). The ladders formed by TC-3-1 were generally shorter than those of TC-1-1 possibly due to its weaker binding force in the elongated direction of the ladder. The AFM images showed that TC-3-2 formed a mixture of large arrays and wrapped tubular shapes (> 400 nm × 2 μm, Fig. 2f and Supplementary Fig. 8), suggesting that the double T junctions design with simple matching rules encouraged large array formation.

The design of TC-4-2 was generated by adding one more helix and crossover on the horizontal arm of TC-3-2 tile. As shown in Fig. 2h, the two perpendicular arms of TC-3-2 were joined by a T-loop and a third DNA helix, without any unpaired nucleotides introduced to the structure. The sticky-end matching rules of TC-4-2 remained the same as that of TC-1-2 (Supplementary Figs. 2 and 10). The native gel results followed similarly with other TC tiles, only showing a major band on top, while the AFM images showed many irregular flat 2D grids and a small number of tubular arrays. The corresponding ladders assembled from TC-4-1 (Fig. 2g and Supplementary Fig. 9) achieved micrometer size (> 1.5 μm). Both the ladder and grid formation demonstrated that

the two-helix arms with increased rigidity were important for DNA tile 2D assembly.

To study the impact of the assembly units, we designed TC-8-3 and TC-9-3 tiles by connecting two TC-1-2 and TC-2-2 tiles with asymmetric sequences (Supplementary Fig. 11). TC-8-3 mainly formed tubular-shaped arrays (Supplementary Fig. 12), while TC-9-3 formed micrometer-sized arrays similar to TC-2-2 (Supplementary Fig. 13). In addition, we also compared the linear assembly by using half-ladder and full-ladder units. Four tiles were designed including TC-8-1, TC-8-2, TC-9-1, and TC-9-2 (Supplementary Figs. 14–18). Based on AFM images, the full-ladder designs, TC-8-2 and TC-9-2, generally formed much longer 1D assemblies than the assemblies with half-ladder tiles. It was possible that the dimer repeating units in the full-ladder design provided stronger and more rigid connection than the half-ladders did, and therefore facilitated the longer linear assembly.

Finally, the sticky-end length is a critical factor for 2D array growth. In TC tile systems, the sticky ends on vertical and horizontal arms need to be investigated individually. We picked TC-8-3 design and shortened its horizontal sticky ends from 4 to 2 nt (Supplementary Fig. 19). After annealing, the 2-nt sticky-end design assembled mostly into flattened 2D array structures or opened tubes, while the original 4nt sticky-end tile formed mainly tubes. Interestingly, after one week of storing in 4 °C, the tubular shapes became the dominant structures in both samples (Supplementary Fig. 19d). Next, we decreased the sticky-end length of the vertical arm of TC-3-2 from 6 to 3 nt (Supplementary Fig. 20a). Under AFM, no large assembly was obtained in the design with 3 nt sticky ends. A possible reason lies in the critical function of the vertical arm in this TC design, in which sticky ends on vertical arms should provide sufficient binding force to maintain the array formation.

## Two-layer and three-layer TC tiles

To demonstrate the versatility of the TC design method, we designed six multi-layered TC tiles including two two-layer (Fig. 3a, b) and four three-layer structures (Fig. 3c–f). The two-layer tiles (TC-5-1 and TC-5-2) were created by overlapping two T-junction helices and connecting them with AX crossovers (Fig. 3a, b and Supplementary Figs. 21–23), and they were different in arm lengths and sticky-end matching rules, resulting in assembling ladders and grids, respectively. Based on the height measurement from AFM images (Supplementary Figs. 7i and 22i), the height of the ladders assembled from two-layer tiles was about two times that of single-layer tiles, confirming the double-layered configuration. The observed tubular arrays formed by TC-5-2 were over 2 μm in length (Supplementary Fig. 23). The measured height of the unwrapped array (-1.9 nm) was about half that of the wrapped tube array (-3.8 nm), validating the tubular array formation (Supplementary Fig. 23i). Although both TC-4-2 and TC-5-2 have a 2-helix DNA bundle as vertical arms, TC-5-2 tended to form a tubular shape (>100 nm × 2 μm) while TC-4-2 did not. The difference lies in the arrangement of the vertical arms and the connection between the vertical and horizontal arms. The vertical arms of TC-4-2 were connected to the same horizontal arm through one joint on the T-loop and one joint on a DNA helix (Supplementary Fig. 24). In comparison, the vertical arms of TC-5-2 bound to the horizontal arms separately using two joints solely on T-loops, resulting in a more flexible structure (Supplementary Fig. 24). Extending the sticky-end length from 4 to 6 nt in TC-5-2 resulted in similar tubular assemblies with some unwrapped tubes (TC-5-3, Supplementary Fig. 20b).

Next, sandwich-styled three-layer tiles were explored by inserting one helix (TC-6-1, TC-6-2) or two helices (TC-7-1, TC-7-2) in between the two horizontal arms (Fig. 3c–f and Supplementary Figs. 24 and 25). The vertical arms of TC-7-1 and TC-7-2 were rigidified by adding crossovers between the two helices. The measured height of a TC-6-1 ladder was -2 nm (Supplementary Fig. 26). The large view of AFM images showed that the ladders formed by TC-6-1 were longer than that of TC-7-1 (Supplementary Figs. 26 and 28). Moreover, both TC-6-2 and TC-7-2 were observed to form micrometer-sized arrays under AFM (Fig. 3d, f and Supplementary Figs. 27 and 29). Interestingly, all the tested TC designs in Figs. 2 and 3 that were intended to create grids were observed to mainly form tubular-shaped arrays after storing at 4 °C for a week (Supplementary Fig. 30).

We demonstrated that the TC design strategy was general and versatile for array construction. All zoom-in AFM images of the TC tiles matched their designed patterns. We concluded that tile structures with a simple topology, proper structural support (strut), robust matching rules, and sufficient binding force and rigidity led to the best self-assembly in TC tile system. Unlike previously reported T-junction tiles that need to be assisted by a substrate for 2D assembly[16,17], our TC tiles were able to form micrometer-sized flat arrays in solution, indicating a great potential of TC tiles for solution-based applications.

## 2-state DNA nanoring reconfiguration

In living systems, reconfiguration plays crucial roles in many biological processes such as enzyme activation, antibody-protein binding, and penetrating regulation[28–30]. Engineering shapeshifting systems has been a long-standing goal. DNA is known as a great biomaterial for constructing dynamic systems due to its high programmability through Watson–Crick base pairing[31]. The systematic exploration of TC tiles suggested that these versatile and resilient structures could emulate the intriguing features of natural assembly, shedding light on the intricacies behind their delicate features. Herein, we developed reconfigurable TC tile-based nanorings to imitate nanopore geometry and their dynamic geometric change, including open-to-close and close-to-open, by enlarging and contracting DNA nanorings.

In order to engineer nanorings of programmable dimensions, a specialized C-shape TC tile was designed, featuring two horizontal arms on a vertical support arm (Supplementary Fig. 31). The assembled nanoring size was determined by the inherent geometry of individual tile, which could be regulated by adjusting the length of the inner and outer horizontal arms. We tested a tile that was designed to accommodate a 12-monomer nanoring (Supplementary Fig. 32). AFM and Transmission Electron Microscopy (TEM) images confirmed the nanoring formation and showed a reasonable assembly yield (-67%, $n = 3230$, n is the total monomer number in counted structures, Supplementary Figs. 32 and 33 and Supplementary Excel). The yield was

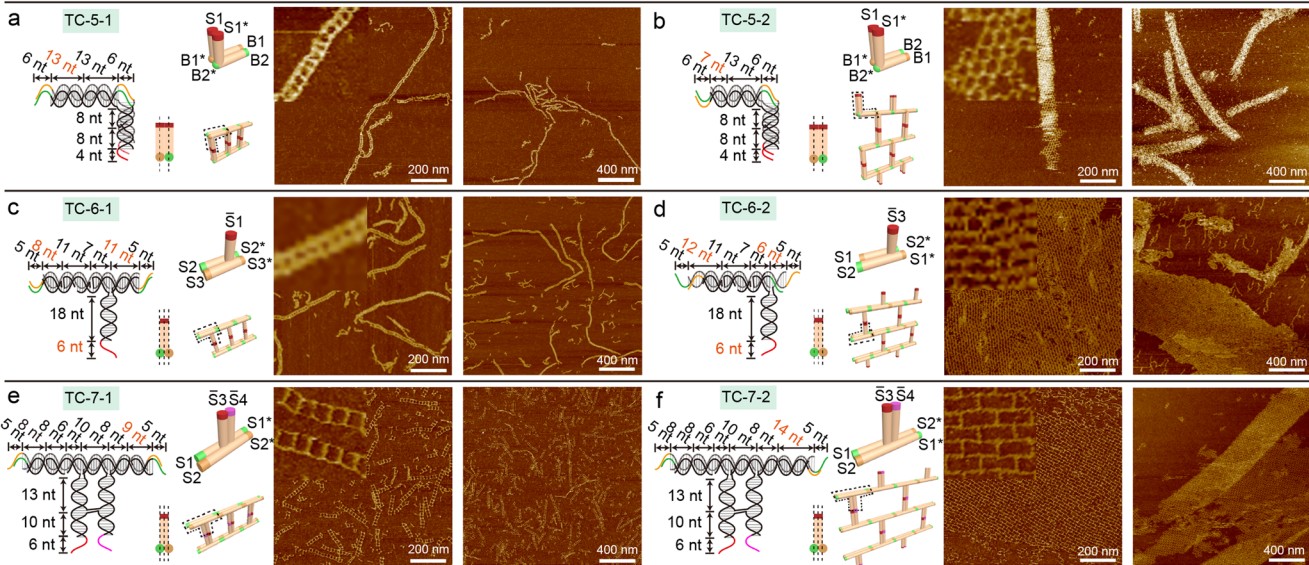

**Fig. 3 | Design schematics, sticky-end matching rules, and corresponding AFM images for two- and three-layer TC tiles. a** TC-5-1; **b** TC-5-2; **c** TC-6-1; **d** TC-6-2; **e** TC-7-1; **f** TC-7-2. Each figure contains the design of tile, cylindrical model of tile (side view and 3D view) and AFM images of assembled arrays. The dash lines in a side view indicate the layers of tiles. The labeling and color coding for sticky-end matching rules are provided in the caption of Fig. 2. All the AFM images are taken after a 66 h annealing process. Inset image size: 100 nm × 100 nm. The inset images are cropped from the corresponding AFM images in this figure. Source data are provided as a Source Data file.

estimated from AFM images and further details of the calculation method were elaborated in Section 1 of the Supplementary information. Functional toeholds were introduced into the horizontal arms of the TC tile to enable the reconfiguration of nanorings by adjusting the lengths of certain arms via adding invader and setting strands (Supplementary Fig. 34)[20]. The diameter of the single nanoring changed according to the horizontal arm changes. As shown in Fig. 3, we designed and studied four pairs of two-state dynamic nanorings, named Pair 1–4, to target the formation of different-sized dynamic nanorings. Each pair of nanorings featured the same monomer number ($N$ value). The calculated diameter difference between the two nanorings were 19 nm, 28 nm, 21 nm, and 33 nm, respectively. AFM images showed that four pairs of two-state nanorings were all successfully assembled. The formation yields of these directly annealed nanorings were estimated from AFM images (details of yield calculation were in section 1 methods of Supplementary information and Supplementary Excel). The two nanorings in Pair 1 ($N = 8$) had the same formation yield as 46% (Fig. 4a, R1, $n = 18175$; R2, $n = 9205$), while the yields were 9% ($n = 9451$) for R1 and 11% ($n = 12435$) for R2 in Pair 2 ($N = 12$) (Fig. 4b). It suggests that the formation yield of the directly annealed nanoring was affected by the number of monomers in a nanoring ($N$). Generally, a higher yield could be obtained when a nanoring needs a smaller number of monomers to be incorporated into the structure. The Z-tile-based nanorings showed similar results:

Pair 3 ($N = 9$) were found to have both well-formed R1 and R2 yield as 35% (Fig. 4c, R1, $n = 12113$; R2, $n = 24589$), while Pair 4 ($N = 14$) showed a yield of 8% ($n = 14864$) for R1 and 15% ($n = 11539$) for R2 (Fig. 4d). In addition, we calculated the yield of reconfiguration for each pair. The transformed R1* & R2* were 43% & 38% in Pair 1, 13% & 8% in Pair 2, 35% & 34% in Pair 3, and 14% & 6% in Pair 4 (Fig. 4a–d, pair 1: R1*, $n = 13259$; R2*, $n = 13868$; pair 2: R1*, $n = 9930$, R2*, $n = 8548$; pair 3: R1*, $n = 20486$, R2*, $n = 17657$; pair 4: R1*, $n = 13567$, R2*, $n = 14510$), respectively. The byproducts observed among the four pairs of nanorings were partially assembled nanorings, opened nanorings, and closed smaller nanorings. A possible factor that affected nanoring formation was an additional nick on the horizontal arm, introduced by the SDR functional structure. The nicked horizontal arm became flexible and thus deformed the overall nanoring configuration.

To have a better understanding of the reconfiguration process, we conducted AFM imaging on the intermediate structures of Pair 1 and Pair 3 nanorings to represent the C-tile and Z-tile-based structures, respectively (Fig. 5a and Supplementary Fig. 35a). Upon the introduction of invader strands to simultaneously displace the inner and outer layer of the nanoring, the intermediates underwent a transformative process to adopt the different curvature of the target nanorings. It was observed that some intermediates maintained the integrity of the inner layer connection, while others broke into curved ladders (Fig. 5b and Supplementary Fig. 35b).

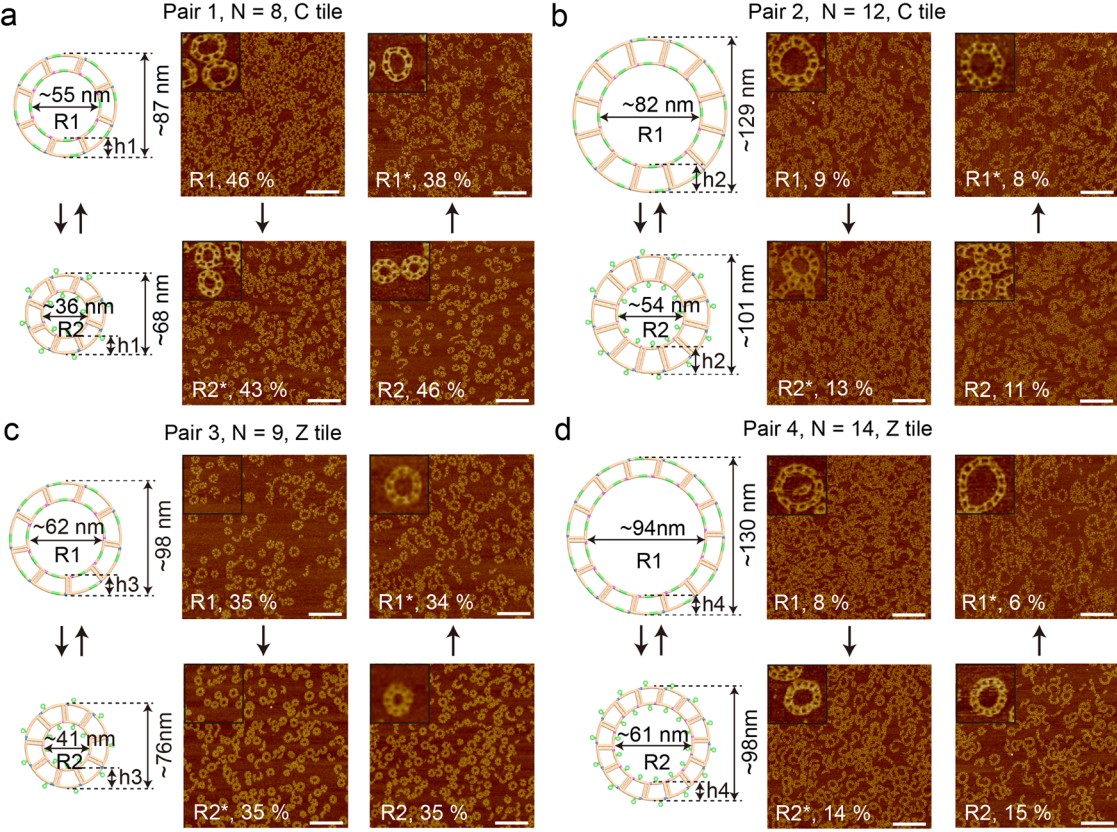

**Fig. 4 | 2-state DNA nanoring transformation. a** Pair 1, C-shape TC tiles, $N = 8$, inner diameter changing between ~55 nm and ~36 nm. **b** Pair 2, C-shape TC tile, $N = 12$, inner diameter changing between ~82 nm and ~54 nm. **c** Pair 3, Z-shape TC tiles, $N = 9$, inner diameter switch between ~62 nm and ~41 nm. **d** Pair 4, Z-shape TC tile, $N = 14$, inner diameter changing between ~94 nm and ~61 nm. The transformation of nanoring is induced by strand displacement reactions. $N$ is the number of monomer tiles required to form a DNA nanoring, and $h$ is the width of the ring. The two nanorings in a pair have the same $N$ and $h$. The only difference is the inner diameter. The estimated diameter difference between the two states of the rings is 19 nm, 28 nm, 21 nm, and 33 nm for pair 1 to 4, respectively. The width of the 2-state nanorings ($h1$ to $h4$) remains the same during reconfigurations. The calculated formation yield of each sample is labeled on AFM images. The counted well-formed nanorings with an $N$ value include $N$-1, $N$, and $N + 1$ structures. The calculation methods are listed in section 1 Methods of the supplementary information. The reconfiguration directions are shown by the arrows. The monomer tiles are marked with orange color. The green regions represent the toehold-mediated strand displacement functional domains. Image size: 1.5 μm × 1.5 μm, scale bar, 300 nm; inset image size: 150 nm × 150 nm. The inset images are cropped from the corresponding AFM images in this figure. Source data are provided as a Source Data file.

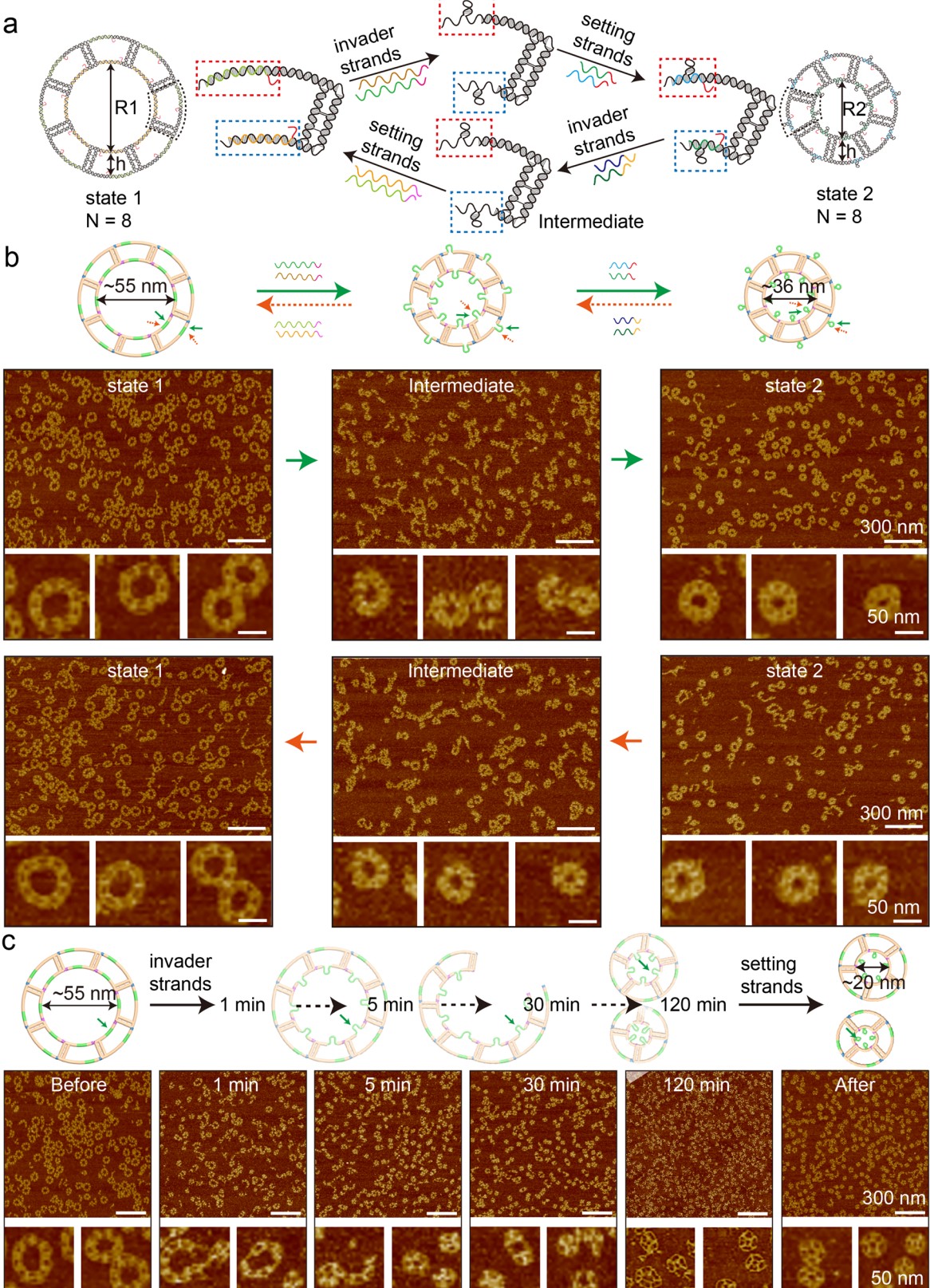

**Fig. 5 | Intermediate structures during transformation for 2-state nanorings.**
**a** Schematics of nanoring transformation induced by strand displacement reactions. From state 1 to state 2, two of invader strands and two of setting strands were added, and vice versa. Invader and setting strands are marked with different colors and toehold regions are marked with pink. **b** The schematics and AFM images of two nanoring transformations, transformed intermediates and products. Here, the C-tile-based nanoring is used as an example (N = 8), and the N is retained after transformation. The typical assemblies are cropped out from the corresponding AFM images in this figure. The reconfiguration directions are indicated by the arrows. Green arrow shows the transformation from state 1 to state 2 and orange represents the reverse direction. **c** Schematics and AFM images showing the transformation of a big nanoring to a small nanoring with a decrease in N value (from N = 8 to N = 5). Samples at different time points are tracked by AFM imaging after adding invader strands, including 1 min, 5 min, 30 min, and 120 min.

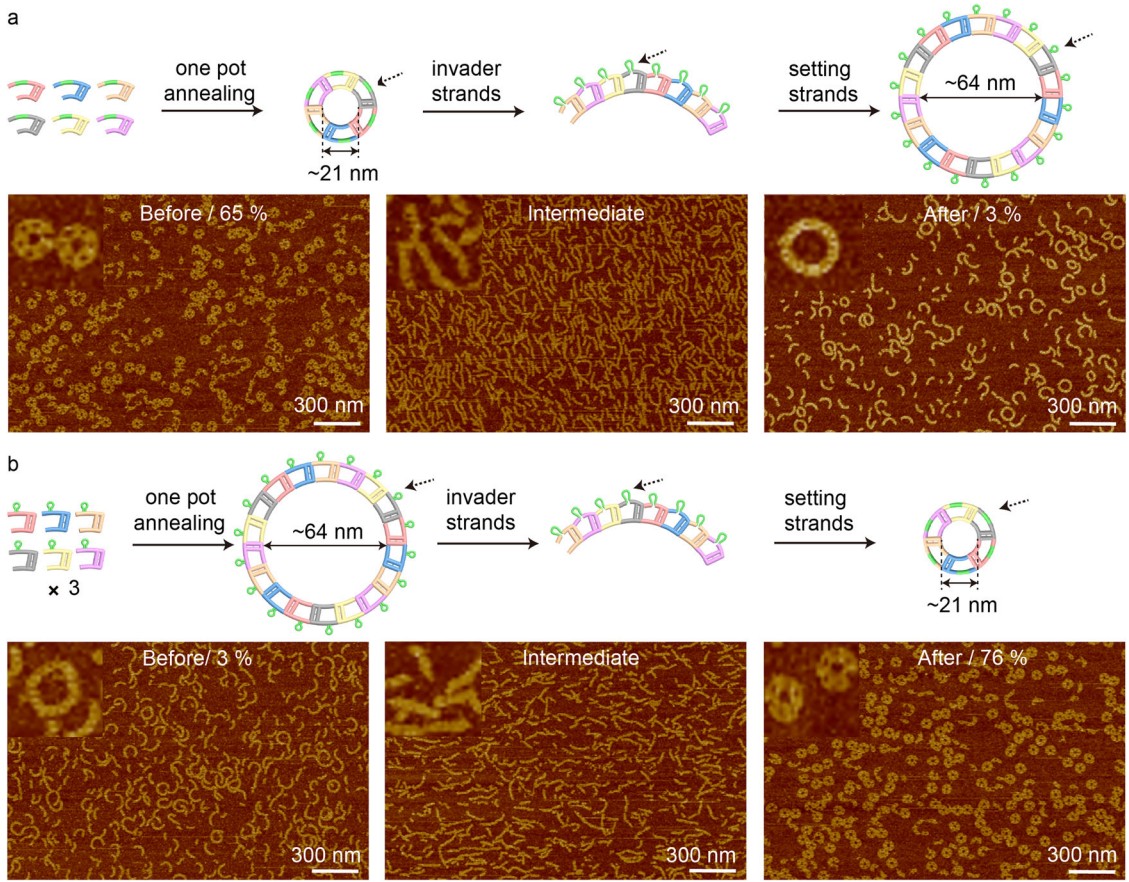

**Fig. 6 | Single-stranded tiles (SST) for precise nanorings and their reconfiguration.** Schematics and AFM images, **a** 6-monomer nanorings assembled from annealing, intermediate ladders, and transformed 18-monomer nanorings; **b** 18-monomer nanorings assembled from annealing, intermediate ladders, and transformed 6-monomer nanorings. The SST nanorings contain six C-shape monomer tiles with different sequences, while the big nanoring contains three times the monomers present in the small nanoring. The six monomers are labeled with six different colors. Green color represents the toehold-mediated strand displacement functional domain. The assembly yield of each sample is labeled on corresponding AFM images. The formation yield is calculated based on the exact $N$ value (see section 1 Methods of the Supplementary Information). The dashed arrows point out the SDR reaction domain. The zoom-in AFM images are cropped from the corresponding images in this figure. Source data are provided as a Source Data file.

Furthermore, we studied the reconfiguration process of the 8-monomer nanorings resizing into 5-monomer nanorings. As shown in Fig. 4c, after adding invader strands to trigger reconfiguration, we directly imaged a series of samples at different time points including 0, 1, 5, 30, and 120 min. Most of the big nanorings started to break into curved ladders after 1 min. Representative intermediate structures were captured, where a curved ladder adopted curvatures half from the large nanoring ($N = 8$) and half from the small nanoring ($N = 5$) (Fig. 5c). In 120 min, most of the 8-monomer nanorings disappeared. After adding setting strands, the original large nanorings were displaced by small nanorings (37%, 21%, 3% for 4-, 5-, and 6-monomer nanorings, details see Supplementary Excel) and curved ladders (details see section 1 in Supplementary methods for yield calculation). In addition, resizing process in the Z-tile-based nanorings were investigated and showed similar results (Supplementary Fig. 35c).

To achieve even finer control over nanoring dimensions, the single-stranded tile (SST) method was employed in our system[32,33]. Based on C-shape TC tiles, we created 6-monomer and 18-monomer nanorings, and further demonstrated their formation and reconfiguration (Fig. 6, Supplementary Fig. 36, and Supplementary Table 14). The SST nanorings were designed with well-defined sizes and addressable surfaces because the nanoring size is determined by both the geometry of each tile and the specific sequence design. For example, six tiles with distinct sequences assembled into a 6-monomer nanoring. Omitting one monomer or adding one more would prevent the creation of a nanoring

due to the matching rules of the sticky-end sequences. The formation yield of 6-monomer and 18-monomer SST nanoring was 65% ($n = 18996$) and 3% ($n = 15516$), respectively (Supplementary Figs. 37–40). The 2-state SST nanoring reconfiguration was achieved by SDR. First, the corresponding invader strands released part of the structures on the outer layer of the 6-monomer nanoring and thus changed the curvature of the nanoring (Fig. 6a and Supplementary Fig. 36). We observed the intermediate structures as curved ladders. Next, adding setting strands led to the formation of 18-monomer nanorings by combining three pieces of 6-monomer ladders. The transformed 18-monomer nanoring had a similar low yield of 3% (Supplementary Fig. 41, $n = 21,168$). The reversed process was conducted to change 18-monomer nanorings to 6-monomer ones (Fig. 6b). Similar intermediate structures were obtained. The transformed 6-monomer SST nanoring had a formation yield of 76% ($n = 15552$), indicating that some unpreferred byproducts may be corrected during SDR process (Supplementary Fig. 42).

In this section, we demonstrated several 2-state reconfigurable nanorings that undergo resizing processes in a highly programmable manner, which would be quite challenging for the classic T-junction tiles[19–21].

## 4-state DNA nanostructures

To demonstrate the robustness and generality of our approach, we applied it to design four sets of 4-state nanostructures with a total of 16 objects with 48 reconfiguration paths (Fig. 7 and Supplementary

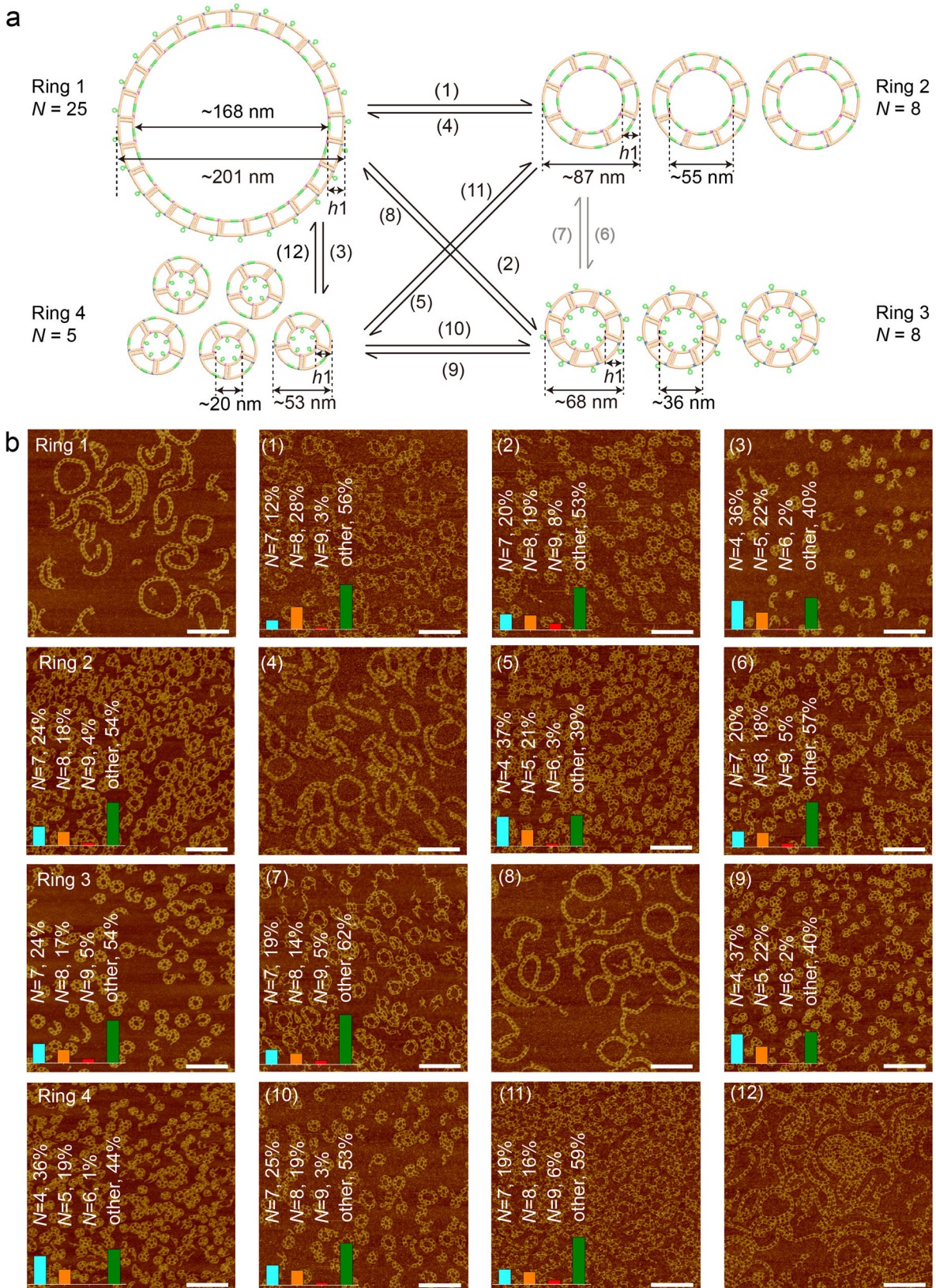

**Fig. 7 | 4-state ring-to-ring reconfiguration based on C-shape TC tiles.**
**a** Schematics of the one-to-one reconfiguration process between four rings, through a total of 12 paths. The four nanorings are Ring 1 ($N=25$), Ring 2 ($N=8$), Ring 3 ($N=8$), and Ring 4 ($N=5$). Each path is labeled with a number from (1) to (12), and the arrow shows the reconfigured direction. The Monomer tile is marked with yellow. Green color represents the toehold-mediated strand displacement functional domain. **b** The AFM images of each nanoring and the reconfiguration product. The first row presents the Ring 1 and its reconfiguration products through

path 1, path 2, and path 3, from left to right, respectively. Similarly, the last three rows depict Ring 2, Ring 3, and Ring 4 and the corresponding three paths of reconfigured nanorings, from top to bottom, respectively. The yield counting information is inserted on top of the AFM images. In the yield analysis graphs, the bars for nanorings with $N-1$, $N$, $N+1$, and other monomers are marked with light blue, yellow, red, and green color, respectively. The information of invader and setting strands of each transition is summarized in Supplementary Table 13. Image size: 1 μm × 1 μm, scale bar: 200 nm. Source data are provided as a Source Data file.

Figs. 43–47). The first set of 4-state structures was designed based on C-tile with horizontal arm lengths as 7&6 turns, which could switch to three different tiles with horizontal arm lengths as 9&6, 7&4, and 9&4 turns (Supplementary Fig. 43). Correspondingly, they formed four switchable nanorings with different monomer numbers and diameters (Fig. 7a). All the possible transformations between the four states were through 12 paths, in which path 6 and path 7 were same as the 2-state nanoring transition in Fig. 3a. The experimental realization of the four states of nanorings was confirmed by AFM imaging (Fig. 7b). We validated each of the 12 reconfiguration paths, producing resized nanorings that were identical to the original design. The intermediate structures for each path were observed under AFM by imaging samples with invader stands (Supplementary Fig. 48). A statistic study was conducted to calculate the formation yield for each starting nanoring and its corresponding transformed nanorings (Fig. 7b and Supplementary Excel). This data aligned with the 2-state reconfiguration systems, in which nanorings with a smaller number of monomers generally had a relatively high yield.

Further, the 4-state reconfiguration system based on Pair 3 Z-tiles was tested (Supplementary Fig. 46a). Interestingly, one reconfiguration of the nanorings formed ladders, where the upper and lower horizontal arms of Z-tile were of the same length. The shapeshifting between the nanorings and the ladders showed that minor changes in an individual tile may accumulate into a dramatic reconfiguration. The AFM images proved that the 4-state structures switched between all configurations (Supplementary Fig. 46b). The intermediate structures for each path were analyzed as well (Supplementary Fig. 49).

In addition, we imaged and analyzed another two sets of 4-state reconfigurations based on the design of the Pair 2 and Pair 4 tiles (Supplementary Figs. 45 and 47). All the corresponding yield counting data were summarized in the Supplementary Information (Methods of Supplementary Information, Supplementary Figs. 50–113, and Supplementary Excel). We concluded that both C-tile and Z-tile could switch between four dynamic states. The shape changes from ring to ring, ring to ladder, or ladder to ring were all achieved through SDR. The multiple structural switching process brings new possibilities for dynamic DNA nanodevice design and potential applications based on it.

To conclude, we developed a series of DNA TC tiles with enhanced rigidity and angle control that enable the self-assembly of large 1D and 2D patterns in solution. We then illustrated the 2-state reconfigurations of SST nanorings and single-monomer nanorings and demonstrated four sets of 4-state systems with effective transformations between ladders and/or rings with pore sizes spanning ~20 nm to ~168 nm. These TC tiles adopt advantages from both AX and T junctions and can be used to achieve various nano-constructions beyond what we have demonstrated here. Overviewing the existing practices in DNA and RNA nanotechnology, TC tiles can be adapted into different design approaches. For instance, multiple-stranded TC tiles can be rerouted to have single-stranded folding pathways in RNA (A-form), allowing the enzymatic mass production of the tiles and even in vivo applications. Another possible future work is to employ the design methodology of TC tiles to form DNA crystals. The two-layered or three-layered TC tiles can assemble into micrometer-sized ladders or grids so that they can be modified to grow in 3D and form designer DNA crystals for the arrangement of functional molecules, such as protein, drugs, and nanoparticles[34–36].

One important advantage of TC tile was their capability to direct various dynamic reconfigurations of nanorings. To date, DNA nanoring has been realized by a few design approaches including the single-stranded DNA, DNA tiles, and DNA origami[17,19,20,37–39]. However, previous tiles generally have more flexible structures and limited geometry of the tile frames that make further modification challenging. In this work, the C- or Z-shape TC tiles have rigid frame arms and angle-controlled T-loops. Therefore, both the upper and lower horizontal arms can easily incorporate SDR functional groups for dynamic reconfigurations. Notably, the TC tile assembled dynamic nanorings have comparable sizes with DNA origami (50–200 nm) but using only ten strands to form the nanoring, while an origami design generally needs hundreds of staple strands. In addition, one cycle of dynamic shapeshifting only needs a few invader and setting strands, which yields a simple and economic system. Overall, the DNA TC tiles for arrays and dynamic nanorings can facilitate various applications in the future, such as sensing with fluorophore and quencher modification, and creating an artificial nanopore with dynamic pore sizes on cell membrane[40–42].

## Methods

### Formation of DNA nanostructure
All the DNA strands were purchased from Integrated DNA technologies Inc. (idtdna.com) at 25 or 100 nmol scale, then purified by denaturing PAGE gel. For DNA nanostructure assembly, equimolar amounts (1 μM) of all the strands in 1× TAE•Mg²⁺ buffer (2 mM EDTA•Na₂ •12H₂O, 40 mM Tris base, 20 mM acetic acid, 12.5 mM (CH₃COO)₂Mg•4H₂O) were annealed in a thermocycler (Eppendorf) with different annealing programs. All the chemicals needed for preparing the buffer were purchased from Sigma-Aldrich. All the annealing programs were set as cooling from 95 to 4 °C with different incubation times (2 h, 16 h, and 66 h). For detailed strand sequences information see section 3 of Supplementary information.

### Sample preparation for SDR
The starting materials of SDR were DNA nanostructures (nanoring or ladder) assembled from C-shape or Z-shape TC tiles that were incubated in PCR tubes using a 2 h quick annealing program (The concentration of all DNA strands is 1 μM). To start the SDR cascade, five times the amount of the invader strands (5 μM) was pipetted into the PCR tube and kept at 25 °C for 2 h to facilitate the strand displacement. Then ten times the amount of the setting strands (10 μM) were added to the system and incubated at 25 °C for 2 h to self-assemble the reconfigured nanostructures. Details can be found in Supplementary Methods.

### Native PAGE
In all, 4.5% native PAGE gel was prepared with Acrylamide/Bis 19:1 and 1× TAE/Mg²⁺ buffer and used for annealed samples characterization. The gel was run in 1 × TAE/Mg²⁺ buffer at 4 °C and stained with 1× gel red by post-staining method, then imaged by Biorad Gel Doc XR Imaging System.

### AFM imaging
All of the samples were prepared as liquid samples and the steps were as follows: (1) 10 μL 1× TAE-Mg²⁺ buffer was added onto a freshly cleaved mica (Ted Pella) surface, (2) 1 μL of sample and 2 μL of NiCl₂ solution (100 mM) were added and incubated for 1 min, (3) 70 μL 1× TAE-Mg buffer was added and incubated for 5 min, (4) the buffer was removed with tissue, (5) 70 μL TAE/Mg²⁺ Buffer and 2 μL of NiCl₂ solution (100 mM) were added before proceeding to AFM imaging. Images were collected on a Bruker Fastcan AFM in the ScanAsyst in liquid mode using ScanAsyst-fluid+ probe (Bruker).

### Statistics and reproducibility
Data quantified in the manuscript comes from single experiments containing hundreds of monomers. Each experimental condition was reproduced at least three or more times for gathering of AFM images, optimizing annealing and imaging protocols, and control experiments.

### Reporting summary
Further information on research design is available in the Nature Portfolio Reporting Summary linked to this article.

## Data availability
The authors declare that the data supporting the findings of this study are available within the paper and its Supplementary Information. Source data for all the statistical analysis of yield are provided with the paper. Raw data for figures with cropped AFM and gel images are supplied in Source Data files. Source data are provided with this paper.

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

## Acknowledgements
This work is supported by a US National Science Foundation (NSF) Faculty Early Career Development Award (DMR-2046835) and a faculty Startup Fund from Rutgers University.

## Author contributions
Q.Y. and F.Z. conceived the idea. Q.Y. and X.C. designed the structures. Q.Y., X.C., and J.L. prepared and characterized samples. Q.Y. and F.Z. performed the data analysis. The manuscript was written through the contributions of all authors. All authors have given approval to the final version of the manuscript.

## Competing interests
The authors declare no competing interests.
