## [Peer Review File · Nature Communications]

DNA T-Shaped Crossover Tiles for 2D Tessellation and Nanoring ReconfigurationREVIEWER COMMENTS

Reviewer #1 (Remarks to the Author):

In this study, Zhang and co-workers introduced DX-T tiles for the periodic arrangement of the motifs into 2D lattices and rings of different diameters. Both static and dynamic structures based on the structural motifs were demonstrated. It is a comprehensive study about a new structural motif and the results are convincing. However, when compared to T-junction introduced earlier (references 17 and 18), the implementations shown in this study about static and dynamic constructs are rather progressive. This reviewer is also a little confused about the scope of the manuscript. Is it about DX-T tiles or is it about the reconfiguration of rings? Maybe the authors can think of dividing the article into two, one about static structures and the other about dynamics. In particular, this reviewer is excited about the dynamic reconfiguration of the rings. According to the results about the structural reconfiguration, a reader can assume an alternative scenario in which a certain ring breaks apart into individual tiles first and then re-assemble into another ring. It would be nice if the authors could present more details about the process. Especially, cell division was provided as a reference point of the reconfiguration process. It would be wonderful to directly observe one ring divided into two (e.g. a ring of eight Z-tiles to be divided into two rings of four Z-tiles).

In general, this reviewer would suggest a major revision to define a clearer scope before re-submission.

Reviewer #2 (Remarks to the Author):

This sizable study by Yang et al describes several designs for DNA tiles based on combining well known double crossover T junction tiles with the goals 1) to create a more rigid building block for 2D tilings and 2) to dynamically control block geometry for DNA ring resizing. This paper has a lot of high quality AFM data, but the motivation, design, results, and data analysis could be significantly improved. The manuscript needs major revisions to become useful for anyone. Hope the following comments can help improve it. Please note that I was reading this paper in the most charitable mindset, since I am all for supporting young new labs and I am a fan of this PI's past work.

Major comments:

The target structures assembled in this work are polydisperse and poorly controlled. It feels like a trip back in time to the DNA nanotech field of 20 years ago. To the times where we could be satisfied with constructing non-precise unbounded polydisperse structures.

Developing new tiles studied in this work is commendable, but they need to provide new capabilities. The field can currently construct very similar to reported architectures with much higher precision, e.g.

<https://www.nature.com/articles/nature24648>

<https://www.nature.com/articles/s41596-020-0331-7>

New tile motifs are still important to develop even if they cannot compete with current capability, but the work should show some advantages to be of general use in the field, e.g.,

<https://pubs.acs.org/doi/full/10.1021/jacs.2c11731>

It is surprising that the authors did not explore variation of the sticky end lengths and sequences. Tubing can be a result of flexibility of the tile rather than intrinsic curvature. The flexibility of the tile is a function of sticky end sequence which defines tile-tile melting point: the higher the melting point, the more flexible the tiles during the formation of the target multitile structure. In addition to adding struts to rigidify tile I would recommend simply shortening the sticky end sequence wherever possible.

On the related note, the authors should analyze the hybridization energy of all the sticky ends used.

Also, it is important that the detailed anneal protocols are reported. Right now, only the temperature range and total time is provided. Temperature protocols and sticky end design have critical effects on the morphology of multifarious assembly studied here.

The design choices need to be logically justified. Some of them appear arbitrary in this manuscript. It's understandable that it's impossible to try many apparent designs, but some

design space choices should be justified. Right now they seem arbitrary. For example, why both ladder and grid versions were studied for tile 10/12, but only ladder versions for tiles 8, 9, 11? Also, no ladder version were studied for Tiles 1-6, but the tile 7.

The motivation for ring reconfiguration study is not convincing at best. Biological nanopore sizes are angstrom-precise to permit just the right ions and molecules. Nanorings with a broad size distribution described here are much less useful. We now have access to precisely controlled size of nanopores that can be gated, e.g., a paper co-authored by the PI of this study. Why go back in time to polydisperse rings? Simple building motif is not strong enough motivation. Besides it would be really hard to make a pore from the described giant ring, not talking about reconfiguring it.

Ring yield calculations are misleading as it seems that the authors report the target nanoring sizes (12 and 13?, not clear!) versus total rings of all sizes and ignore the assemblies that are not rings, while saying something different from what they did in the caption of Fig S18. The authors did not consider the possibility that their structures are actually mostly helical staircases rather than rings. It is especially likely for the 8 and 16 h anneals (those staircases are likely broken up partially to give the rings on mica surface). TEM would be useful here.

The SI figures should provide larger view areas 10umx10um in high resolution. Current images are two small areas to get a statistically sound idea of global morphology of the samples. For example, I am not convinced that tile 12 does not form tubes. All the images provided indicate that it does form tubes, they are just large in diameter compared to the other more flexible tiles. A larger area (10x10 micron) image will be more conclusive.

What is reported as ladders for tiles 7 and 10 actually looks more like rings that are flattened on mica during AFM imaging. TEM and especially cryoTEM would help resolve

these ambiguous results.

Fig 1b is misleading in the way it represents ring reconfigurations as giving precise sizes. Experimental results show size distributions. The figure should be corrected to reflect that, e.g., by showing multiple sizes of the rings for each state.

Minor comments:

It would be nice to make more precise (not polydisperse) and addressable (not periodic) assemblies from any single motifs described.

It would also be nice to explore combining different motifs together in one pot or multistage assembly.

Why do you call it “SELF-reconfiguration of DNA nanoring”. It’s just a reconfiguration as an external strand is added.

This work is relevant and should be cited

<https://pubs.acs.org/doi/10.1021/acsbio.2c01176>

Reviewer #3 (Remarks to the Author):

This manuscript reports the design and characterization of DNA nanostructures that assemble into lattices and rings. The main novelty of the design is that it takes inspiration from DNA tiles and T-junctions to generate a new type of building block that is more versatile than plain tiles or T-junctions alone. The authors demonstrate a number of tile variants and characterize them through AFM and gel electrophoresis. They also use strand displacement to show how the building blocks can be conformationally switched, resulting

in expansion or contraction of ring diameter.

The work seems sound and of interest to the DNA nanotechnology community, however I note several aspects that need improvement. Many of the experiments need to be described more carefully and clearly, reporting their statistics. The experiments that I found to be the most interesting (reconfiguration of rings), are not explained well and I can't evaluate their significance.

Detailed comments are below.

1. In my opinion a deeper motivation for this work is needed. The technical novelty is clear, but what can these structures achieve that isn't achievable with other methods? In the conclusion, the authors mention advantages such as rigidity, reduced number of strands when compared to origami, and possibility of dynamic shape shifting through DNA strand displacement. To me these aspects seem legitimate when discussing rings, but the part of the paper focusing on rings needs improvement as I will note below. For lattices, I am not sure these advantages are major, since DNA lattices based on tiles in various flavors have been demonstrated for almost 20 years, including their assembly and disassembly (see related comment below).

2. There is no discussion about statistics of the reported measurements. The authors need to report whether experiments were replicated, how many AFM images were analyzed, and how many rings/ladders were considered when reporting their features (mean number of monomers, diameter etc), which at the moment also lack standard deviation.

For figures 4, 5, and 6 to be convincing, the authors should report bar charts (or histograms), in which statistics are clear for each state (how many rings were measured, mean and standard deviation).

3. The references regarding strand displacement on tile-based DNA structures needs to be improved. A few that I think should be cited are:

Feng, Liping, et al. "A two-state DNA lattice switched by DNA nanoactuator." *Angewandte Chemie* 115.36 (2003): 4478-4482.

Zhang, David Yu, et al. "Integrating DNA strand-displacement circuitry with DNA tile self-assembly." *Nature communications* 4.1 (2013): 1965.

Green, Leopold N., et al. "Autonomous dynamic control of DNA nanostructure self-assembly." *Nature chemistry* 11.6 (2019): 510-520.

4. I am confused by the choice of nomenclature throughout the paper. In Fig. 1a, middle, I do not understand why any of the monomers on the top row can be called DX, which stands for double crossover. I only see one crossover in most of them. For the 2 and 3 layer tiles, it is not clear what are the interactions of the helices in parallel laying on the plane perpendicular to the page. Are there any crossovers there?

In Fig. 1b, left, this tile has 3 crossovers in the vertical arm, and only one strand crosses over at each point. Again, I am confused why this is called a DX tile.

I am also very confused by what are 2-layer vs 3-layer tiles. The 1 layer is its own class, since in that case all helices appear stacked on on the xy plane. But when we go to the 2-3 layer, what is the number (2 or 3) referring to, and with respect to what are we layering? Please add a paragraph where this is clarified.

5. In Fig. 2, I am confused by the order of presentation of the tiles. In my opinion, it would make sense to present them in order of complexity, with Tiles 3, 5, and 6 going first, and later Tiles 1 (which includes 2 monomers of Tile 3 arranged to be symmetric relative to the y axis) and Tile 2 (which includes 2 connected Tile 4). The authors themselves acknowledge that the structures formed by Tiles 1 and 2 are identical to those achieved by Tiles 3 and 4 (which appear to form larger lattices). It would help to clarify why two approaches were pursued to assemble the same structure.

I also note that the authors consider the presence of tubular assemblies (for Tiles 1 and 5 in particular) to be a nuisance, whereas I think it's a feature of their design that would be interesting to explore.

6. In Fig. 3 - Here I realized that the color coding of domains in the second row of schematics is inconsistent, then I looked back at Fig. 1 and realized they are inconsistent even there.

E.g. what is called domain 1 is in dark blue in Tiles 7, 8, 11, then light blue in Tiles 10 and 12. Colors are inconsistent also for the other domains 2 and 3. Perhaps this doesn't matter, but it is confusing for the readers.

Tile 7 appears identical to Tile 5 in Fig. 2. The authors say there is a different sticky end matching rule, but I could not figure out what it is based on the domain color/nomenclature used here.

Finally, why is Tile 7 considered 2-layer but Tile 5 is not? This goes back to the nomenclature confusion I mentioned at point 4.

7. Fig. 5 and 6: Which DNA strands were added to achieve each state transition?

For Fig. 4, we have Fig. S19 to clarify which strands were added. Something similar is needed for Figs 5 and 6.

Strands for Fig. 5 and 6 do not seem to be reported in any SI table either. If the experiments in Fig. 5 and 6 were done recombining strands used for previous figures, it needs to be made clear how.

The criterion for adding a number to each arrow of the pathway in Figs 5 and 6 does not seem to follow any logic nor match with numbering in Fig. 4, and the names of the SDR strands reported in SI table 4-5-6 (which refer to Fig. 4 anyway) does not closely follow the numbering in the main paper.

For the state changes described in Fig 6, the authors note that there is a change in the number of monomers present in a ring:

“Due to the change of monomer number in each ring, individual Ring 1 split into three of Ring 2 (or Ring 3) or five of Ring 4, showing an intriguing self-division process.”

I cannot agree with this statement given the poor description available.

I am not even sure the sentence makes sense - do the authors want to say that Ring 1 splits into 3 or 4 other rings because of the change of monomers in a ring? The reason for diameter change should be the addition of invader/settling strands, that then causes the change in monomer number.

Anyway: there is no assessment of whether the rings are splitting, or rather they are disassembling into monomers, which are then conformationally switched and allowed to reassemble. So I think it is misleading for readers to claim that the rings are splitting.

Minor:

There is often a mention of “vibration” of arms, but it is not clear in which direction/plane this is happening, and I didn’t find this explanation helpful.

The authors need to explain how the yield was calculated and over how many experiments in the main paper. I found an explanation of how yield was computed in the caption of SI Fig. 18. It’s not clear whether the classification of well-formed or misformed ring was done by hand or through an image processing software.

The figures in the main paper show up ok in the pdf but were small and very pixelated when printed. In general I found the tile diagrams in the SI much easier to understand as one can follow the backbone of each individual strand marked in a different color.

A consistently annoying thing in the SI is that captions of a figure are often in the next page.

I found a few grammatical errors and expressions that can be improved. Please proofread and check the text.

REVIEWER COMMENTS

Reviewer #1 (Remarks to the Author):

1-1. In this study, Zhang and co-workers introduced DX-T tiles for the periodic arrangement of the motifs into 2D lattices and rings of different diameters. Both static and dynamic structures based on the structural motifs were demonstrated. It is a comprehensive study about a new structural motif and the results are convincing. However, when compared to T-junction introduced earlier (references 17 and 18), the implementations shown in this study about static and dynamic constructs are rather progressive.

Response: We appreciate the reviewer's positive comments on the comprehensiveness of our study. We would be happy to highlight the novelty of this study, particularly compared to previous works.

The assembly of the T-junction was first introduced in reference 17. Compared to this initial effort, our T-shaped crossover (TC) tiles (DX-T tiles in previous version) have an important advance in the 2D array assembly. The T-junction in reference 17 can only assemble into large arrays with the help of a substrate (e.g., a mica surface) due to the unwanted flexibility and instability of the original tile design. While in our current work, we achieved the in-solution assembly of TC tiles without the need for a substrate and obtained a series of 2D patterns by employing a crossover-reinforced design strategy. To our knowledge, this is the first report to enable micrometer-sized array assembly in solution based on T-junction.

In reference 18, the authors used one T-junction design, which was introduced in reference 17, to form a nanoring. Then they used this nanoring as a template to direct the assembly of a second layer of ring by supplying individual tiles. The final products included the original template nanorings (parent) and the newly assembled large nanorings (children). In our current system, we described reconfigurations of nanorings, where the starting nanorings were able to change themselves into new shapes or sizes. Therefore, we demonstrated a new dynamic feature in our TC tile system compared to reference 18.

In addition, we also added the single-stranded tiles (SST) self-assembled nanorings in the revised manuscript (Fig 5). These SST rings have well-defined sizes with addressable surfaces.

In summary, this TC tile system has shown its unique advantages in both static and dynamic assemblies, providing a versatile design route for creating DNA-based materials.

1-2. This reviewer is also a little confused about the scope of the manuscript. Is it about DX-T tiles or is it about the reconfiguration of rings? Maybe the authors can think of dividing the article into two, one about static structure and the other about dynamics.

Response: We thank the reviewer for the suggestion. During the drafting process of this manuscript, we indeed pondered over the appropriate approach: whether to present it as one comprehensive paper or divide it into two distinct papers. In the current manuscript, our rationale revolves around a systematic study of the TC tile design methodologies, followed by an exploration of dynamic self-assembly. This comprehensive study of TC tile system will serve as a valuable resource for advancing future DNA-based materials design.

1-3. In particular, this reviewer is excited about the dynamic reconfiguration of the rings. According to the results about the structural reconfiguration, a reader can assume an alternative scenario in which a certain ring breaks apart into individual tiles first and then re-assemble into another ring. It would be nice if the authors could present more details about the process. Especially, cell

division was provided as a reference point of the reconfiguration process. It would be wonderful to directly observe one ring divided into two (e.g. a ring of eight Z-tiles to be divided into two rings of four Z-tiles). In general, this reviewer would suggest a major revision to define a clearer scope before re-submission.

Response: The reviewer's interest and suggestion are appreciated very much. Here, we have studied a series of reconfiguration processes by imaging intermediate structures during strand displacement reactions and summarized the new data into the revised manuscript (Fig 4). In total, 26 intermediate structures were observed. Here, we only included 3 sets of reconfiguration processes and highlighted some of our data here.

Figure R1. Schematics and AFM images of nanoring (N=8) reconfigurations and two intermediates.

First, we tested the reconfiguration of an 8-monomer nanoring changing sizes between 55 nm (state 1) and 36 nm (state 2). After adding invader strands, we incubated the sample at 30 °C for 120 minutes and then directly deposited it on mica for AFM imaging. In both reconfiguration reactions, we observed the intermediate structures as a mix of flexible nanorings and curved ladders (Fig. R1).

Figure R2. Schematics and AFM images of a nanoring (N=8) resizing into small nanorings (N=5). After adding invader strands, the AFM images of intermediates are taken at 1, 5, 30, and 120 min.

In the second set of reconfigurations, we studied the 8-monomer nanorings resized into 5-monomer nanorings. After adding invader strands to trigger reconfiguration, we directly imaged a series of samples at different time points including 0, 1, 5, 30, and 120 minutes. As shown in Figure R2, most of the nanorings started to break into curved ladders after 1 minute. Smaller nanorings ($N=4$ or 5) can be observed even at 1 minute. We observed an intermediate structure, where a curved ladder adopted curvatures half from the large nanoring ($N=8$) and half from the small nanoring ($N=5$) (left zoom-in image in 5 min). In 120 minutes, most of the 8-monomer nanorings disappeared and smaller nanorings were obtained. After adding setting strands, the 8-monomer nanorings were displaced by small nanorings and byproducts structures.

Figure R3. Nanorings from single-stranded tiles (SST) and their transformation. a) 6-monomer SST nanorings opened up to form ladders and finally 18-monomer nanorings. b) 18-monomer SST nanorings opened up to form ladders and finally 6-monomer nanorings.

The third set of reconfigurations were performed on SST systems (Fig. R3). In this reconfiguration system, six tiles with distinct sequences were assembled into a 6-monomer nanoring. During the process, the invader strands were added and released part of the structures on the outer layer of the nanoring and thus changed the curvature of the nanoring. We observed the intermediate structures as curved ladders. After adding setting strands, 18-monomer nanorings formed by combining 3 pieces of 6-monomer ladders. The reversed process can be done through strand displacement reactions as well. Similar intermediate structures were obtained.

In summary, the nanorings can either maintain the inner layer connection or break open to adopt a different curvature for the target new nanorings. We did not observe any alternative scenario as “a certain ring breaks apart into individual tiles first and then re-assemble into another ring.”

Reviewer #2 (Remarks to the Author):

This sizable study by Yang et al describes several designs for DNA tiles based on combining well known double crossover T junction tiles with the goals 1) to create a more rigid building block for 2D tilings and 2) to dynamically control block geometry for DNA ring resizing. This paper has a lot of high quality AFM data, but the motivation, design, results, and data analysis could be significantly improved. The manuscript needs major revisions to become useful for anyone. Hope the following comments can help improve it. Please note that I was reading this paper in the most charitable mindset, since I am all for supporting young new labs and I am a fan of this PI's past work.

Major comments:

2-1. The target structures assembled in this work are polydisperse and poorly controlled. It feels like a trip back in time to the DNA nanotech field of 20 years ago. To the times where we could be satisfied with constructing non-precise unbounded polydisperse structures. Developing new tiles studied in this work is commendable, but they need to provide new capabilities. The field can currently construct very similar to reported architectures with much higher precision, e.g. <https://www.nature.com/articles/nature24648> (single strand DNA origami, teddy bear) <https://www.nature.com/articles/s41596-020-0331-7> (origami nanopore). New tile motifs are still important to develop even if they cannot compete with current capability, but the work should show some advantages to be of general use in the field, e.g., <https://pubs.acs.org/doi/full/10.1021/jacs.2c11731> (bryan wei, Mesojunction-Based Design Paradigm of Structural DNA Nanotechnology)

Response: We appreciate the reviewer's viewpoint. As suggested, we studied and added the single-stranded tiles (SST) self-assembled nanorings that have defined sizes with fully addressable structures. Thanks to the reviewer's constructive suggestions, we added the data of the SST self-assembled nanorings in the revised manuscript (Fig 5 or Fig R3).

We designed a nanoring with a 21 nm inner diameter based on 6 monomer C tiles with distinctive sequences and sticky-end matching rules. In Figure R3a, the six monomers are labeled with six different colors; green represents the toehold-mediated strand displacement functional domain. We first tested the formation yield of this SST nanoring from direct one-pot annealing. The AFM images revealed a high formation yield (65%) of the well-formed 6-monomer nanorings. The reconfiguration of the 6-monomer nanorings was designed to change the 6-monomer nanorings into a larger nanoring with 18 monomers. We studied the intermediate structures by using AFM. After adding invader strands, the 6-monomer nanorings popped open into curved ladders to adopt the new curvature. The setting strands defined the final 18-monomer nanoring structures with a formation yield of 3 %. The reverse process from 18- to 6-monomer nanorings was also investigated in Fig R3b.

2-2: It is surprising that the authors did not explore variation of the sticky end lengths and sequences. Tubing can be a result of flexibility of the tile rather than intrinsic curvature. The flexibility of the tile is a function of sticky end sequence which defines tile-tile melting point: the higher the melting point, the more flexible the tiles during the formation of the target multitile structure. In addition to adding struts to rigidify tile I would recommend simply shortening the sticky end sequence wherever possible.

Response: We thank the reviewer for the suggestion. To study the impact of the sticky end lengths, we selected three tiles, which generated the most tubular assemblies in their original

designs. These three tiles are discussed below, and related data were organized into the current manuscript accordingly.

Figure R4. Impact of sticky end length on the horizontal arms. a) TC-8-3 with 4nt sticky end (orange). b) TC-8-4 with 2nt sticky end (orange). c) AFM images of TC-8-3 right after annealing. d) AFM images of TC-8-4 right after annealing. e) AFM images of TC-8-3 after one week in 4 °C; e) AFM images of TC-8-4 after one week in 4 °C.

We first picked one simple design, TC-8-3, (Fig R4) and shortened the horizontal sticky ends from 4 to 2 nt (highlighted in orange, Fig R4b). After annealing, the 2 nt sticky end design assembled mostly into flattened 2D array structures or opened tubes (Fig R4d), while the original 4nt sticky end tile formed tubes (Fig R4c). Interestingly, after one week of storing in 4 °C, the dominate structures became tubular structures in both designs (Fig R4e-f).

Figure R5. Impact of sticky end length on the vertical arms. a) TC-3-2 with 6nt sticky end (orange) on the vertical arms. b) TC-3-3 with 3nt sticky end (orange) on the vertical arms.

Next, we decreased sticky end length on the vertical arm of original Tile 5 (TC-3-2 in the revised manuscript) from 6 to 3 nt (Fig R5). We didn't observe large assemblies of the tiles with 3nt sticky ends under AFM. The possible reason lies in the critical function of the vertical arm in this design. Sticky ends on vertical arms should have sufficient binding force to maintain the array formation.

Figure R6. Impact of sticky end length on the vertical arms. a) TC-5-2 with 4nt sticky end (orange) on the vertical arms. b) TC-5-3 with 6nt sticky end (orange) on the vertical arms.

In addition, we increased two sticky ends of Tile 11 (TC-5-2 in the revised manuscript) from 4 nt to 6 nt. As shown in Figure R6, both designs formed the target pattern with unwrapped and wrapped tubular arrays.

2-3: On the related note, the authors should analyze the hybridization energy of all the sticky ends used.

Response: The hybridization energy of all the sticky ends has been listed in table 2-13.

2-4: Also, it is important that the detailed anneal protocols are reported. Right now, only the temperature range and total time is provided. Temperature protocols and sticky end design have critical effects on the morphology of multifarious assembly studied here.

Response: We added the detailed anneal protocols in section 1 the methods of supporting information.

2-5: The design choices need to be logically justified. Some of them appear arbitrary in this manuscript. It's understandable that it's impossible to try many apparent designs, but some design space choices should be justified. Right now they seem arbitrary. For example, why both ladder and grid versions were studied for tile 10/12, but only ladder versions for tiles 8, 9, 11? Also, no ladder version were studied for Tiles 1-6, but the tile 7.

Response: Thanks for the suggestion. In our revised manuscript, we added the ladder versions of the original Tile 1-6. To do a comprehensive study, we redesigned several of the tiles and tested both ladder and array formations for every tile design in the previous Figure 2 and 3. In the revised manuscript, we rearranged the figures by moving original Tile 1 and 2 to supplementary information and combining the previous Figure 2 and 3 into one figure. We also improved the logical flow to discuss these tile designs in the revised manuscript (please refer to the 'Results and discussion' section of the manuscript).

2-6. The motivation for ring reconfiguration study is not convincing at best. Biological nanopore sizes are angstrom-precise to permit just the right ions and molecules. Nanorings with a broad size distribution described here are much less useful. We now have access to precisely controlled size of nanopores that can be gated, e.g., a paper co-authored by the PI of this study. Why go back in time to polydisperse rings? Simple building motif is not strong enough motivation. Besides it would be really hard to make a pore from the described giant ring, not talking about reconfiguring it.

Response: We appreciated the suggestion of improving the motivation discussion of this work. We agree with the reviewer that nanopores with precisely controlled sizes are important. In our TC tile assembly system, the nanoring size is determined by the geometry of individual monomer tile by controlling the length of the inner and outer horizontal arms. To understand the size distribution of the assembled products, we conducted a comprehensive statistic study to show the formation of the target nanorings (Fig 6, and Supplementary Fig 42-112). In total, we counted 53 tile designs and calculated the formation yield based on more than 260 structures for each design (please refer to 'Response to question 2-7' to find the yield calculation method). Generally, smaller nanorings ($N \leq 9$) have a reasonable formation yield.

To control the nanoring size more precisely, the single-stranded tile (SST) method was employed in our system. We designed and demonstrated the formation and reconfiguration of a 6-monomer nanoring and an 18-monomer nanoring (Fig R3 or Fig 5). The SST nanorings were designed with

well-defined sizes and addressable surfaces because the nanoring size was determined by both the geometry of each tile and the specific sequence design. For example, in the 6-monomer nanoring design, 5 or 7 monomers cannot close into a nanoring because of the sticky end sequences. The one pot annealing yield of 6-monomer SST nanoring and 18-monomer SST nanoring was 65 % and 3 % (2 h annealing program, Fig R3 or Fig 5), respectively. The 2-state SST nanoring reconfiguration was achieved by toehold-mediated strand displacement reaction (SDR). The invader strands released part of the structures on the outer layer of the nanoring and thus changed the curvature of the nanoring. We observed the intermediate structures as curved ladders. After adding setting strands, 18-monomer nanorings formed by combining 3 pieces of 6-monomer ladders. The transformed small SST nanoring has a formation yield of 76 %, indicating that some unpreferred byproducts may be corrected during SDR process. The reversed process was demonstrated through strand displacement reactions as well. Similar intermediate structures were obtained. However, the transformed big SST nanoring has a low yield of 3%.

Furthermore, in comparison to the gated nanopore paper that we coauthored, the main differences are the complexity and the cost. The gated nanopore was assembled by using DNA origami technique. It involved more than 200 unique DNA strands and more than 14k bases in the structure. For 25 nmole DNA materials only (without counting into the cost of modified DNA that for lipid insertion), the cost of DNA in an origami ranges from \$500 to \$1,000, while our tile design costs much less, ranging from \$50 to \$150. More importantly, the bulky origami structure was very difficult to insert into lipid due to the large surface of DNA origami to be contacted with lipids. It needed 96 cholesterol modification on the surface (32 probes to bind DNA strands modified with 3 cholesterol molecules) [1]. Studies showed that smaller and simpler structures are much easier to be inserted [2-4] with less modification. This is also an important motivation for us to explore DNA tile-based nanoring systems.

2-7. Ring yield calculations are misleading as it seems that the authors report the target nanoring sizes (12 and 13?, not clear!) versus total rings of all sizes and ignore the assemblies that are not rings, while saying something different from what they did in the caption of Fig S18.

Response: Thank you for pointing this out. We corrected the misleading description in the main texts and the caption of Supplementary Fig 32 (Fig S18 in the previous version). Here is an example to show the yield calculations of 12-monomer rings:

$$yield_{N=12} = \frac{\text{number of rings } (N = 12) \times 12 \text{ tiles}}{\text{Total number of } C \text{ tiles}}$$

In this equation, the *number of rings* ($N=12$) was counted from AFM images. The *Total number of C tiles* was estimated from the sum of monomer tiles in any sized rings and unassembled fragments. Detailed yield calculations were added in section 1 the methods of supplementary information.

2-8. The authors did not consider the possibility that their structures are actually mostly helical staircases rather than rings. It is especially likely for the 8 and 16 h anneals (those staircases are likely broken up partially to give the rings on mica surface). TEM would be useful here.

Response: Thanks for the suggestion. As shown in Fig R7, we collected the TEM images of 8 h and 16 h annealed TC-C-6-5.5-3 samples (the C tile in original Fig. S18). Samples were deposited on carbon coated copper grid, then stained by 0.75 % uranyl format with 25 mM NaOH for 1 minute and dried overnight. The results showed that both 8 h and 16 h anneals generated mainly a mixture of rings and ladders. Only a few helical staircases were observed under TEM.

Figure R7. TEM images of the TC-C-6-5.5-3 sample. a) 8 h annealing program; b) 16 h annealing program.

2-9. The SI figures should provide larger view areas 10umx10um in high resolution. Current images are two small areas to get a statistically sound idea of global morphology of the samples. For example, I am not convinced that tile 12 does not form tubes. All the images provided indicate that it does form tubes, they are just large in diameter compared to the other more flexible tiles. A larger area (10x10 micron) image will be more conclusive.

Response: We appreciate this suggestion of the reviewer. We annealed each tile and collected larger view AFM images with 10umx10um in high resolution for all tile designs. Corresponding data were added in supplementary information (Supplementary Fig 1-30). Notably, after storing one week in 4 °C, all the 2D arrays formed tubular shape.

2-10. What is reported as ladders for tiles 7 and 10 actually looks more like rings that are flattened on mica during AFM imaging. TEM and especially cryoTEM would help resolve these ambiguous results.

Response: The TEM images of tile 7 (TC-4-1 in the revised manuscript) and tile 10 (TC-7-1 in the revised manuscript) were obtained after staining samples with 0.75 % uranyl format with 25 mM NaOH. TEM confirmed that both tile 7 and 10 formed short ladders instead of rings (Fig R8). The related experimental data was added in Supplementary Fig. 9f &28j.

Figure R8. TEM images of the ladders formed from TC-4-1 and TC-7-1. a) 8 h annealing program; b) 16 h annealing program.

2-11. Fig 1b is misleading in the way it represents ring reconfigurations as giving precise sizes. Experimental results show size distributions. The figure should be corrected to reflect that, e.g., by showing multiple sizes of the rings for each state.

Response: Thanks for the suggestion. We changed schematics in Fig 1b by adding multiple sizes of the rings for each state.

Minor comments:

2-12. It would be nice to make more precise (not polydisperse) and addressable (not periodic) assemblies from any single motifs described.

Response: We thank the reviewer for providing this constructive comment. We have designed and assembled SST nanorings with defined sizes and addressable structures. The corresponding experimental data was added in the revised manuscript (Fig 5) and Supplementary Fig. 36-41.

2-13. It would also be nice to explore combining different motifs together in one pot or multistage assembly.

Response: As we discussed above, we designed and assembled 6- and 18-monomer nanorings by using 6 different motifs. Both rings were assembled using one-pot annealing. The design and experimental data were added to the revised manuscript (Fig 5) and Supplementary Fig.36-41.

2-14. Why do you call it “SELF-reconfiguration of DNA nanoring”. It’s just a reconfiguration as an external strand is added.

Response: Thanks for pointing this out. In the early version of our manuscript, we used ‘SELF-reconfiguration of DNA nanoring’ to emphasize the resizing process between two states of nanorings with the same number of monomers, without involving any additional tile monomers. We agree with the reviewer that this may be confusing and misleading for readers. In the revised manuscript, we changed this subtitle into ‘2-state DNA nanoring reconfiguration’.

2-15. This work is relevant and should be cited:

<https://pubs.acs.org/doi/10.1021/acsbio.2c01176>

Response: Thanks for your kind reminder. We cited this paper as reference 18 in the revised manuscript.

Reviewer #3 (Remarks to the Author):

This manuscript reports the design and characterization of DNA nanostructures that assemble into lattices and rings. The main novelty of the design is that it takes inspiration from DNA tiles and T-junctions to generate a new type of building block that is more versatile than plain tiles or T-junctions alone. The authors demonstrate a number of tile variants and characterize them through AFM and gel electrophoresis. They also use strand displacement to show how the building blocks can be conformationally switched, resulting in expansion or contraction of ring diameter.

The work seems sound and of interest to the DNA nanotechnology community, however I note several aspects that need improvement. Many of the experiments need to be described more carefully and clearly, reporting their statistics. The experiments that I found to be the most interesting (reconfiguration of rings), are not explained well and I can’t evaluate their significance.

Detailed comments are below.

3-1. In my opinion a deeper motivation for this work is needed. The technical novelty is clear, but what can these structures achieve that isn’t achievable with other methods? In conclusion, the authors mention advantages such as rigidity, reduced number of strands when compared to origami, and possibility of dynamic shape shifting through DNA strand displacement. To me these aspects seem legitimate when discussing rings, but the part of the paper focusing on rings needs improvement as I will note below. For lattices, I am not sure these advantages are major, since DNA lattices based on tiles in various flavors have been demonstrated for almost 20 years, including their assembly and disassembly (see related comment below).

Response: We thank the reviewer’s comments and agree with the reviewer that DNA lattices based on tiles have been developed for 20 years. However, the types of DNA tiles that can be used for 2D assembly are quite limited, including DX tile, multi-arm junction, and SST. Particularly, T junction, which was first introduced in 2009 (reference 17), were not able to assemble into large 2D array in solution. Compared to this initial effort, our T-shaped crossover (TC) tiles (DX-T tiles in the previous version of our manuscript) have an important advance in the 2D array assembly. The T-junction in reference 17 can only assemble with the help of a substrate (e.g., a mica surface) due to the unwanted flexibility and instability of the original tile design. While in our current work, we achieved the in-solution assembly of TC tiles and obtained a series of 2D patterns by employing a crossover-reinforced design strategy. We also systematically studied the factors that

impact the assembly. To our knowledge, this is the first report to enable micron-sized array assembly in solution based on T-junction.

3-2. There is no discussion about statistics of the reported measurements. The authors need to report whether experiments were replicated, how many AFM images were analyzed, and how many rings/ladders were considered when reporting their features (mean number of monomers, diameter etc), which at the moment also lack standard deviation. For figures 4, 5, and 6 to be convincing, the authors should report bar charts (or histograms), in which statistics are clear for each state (how many rings were measured, mean and standard deviation).

Response: We thank the reviewer for providing the suggestions. Each of the TC tiles self-assembly was repeated at least three times. We reported that in the 'Methods' section of the revised manuscript.

For the yield counting, we used four of 3 μm \times 3 μm sized AFM images for each sample. Supplementary Fig 32 shows how to analyze AFM images. Here is an example to show the yield calculations of 12-monomer rings:

$$\text{yield}_{N=12} = \frac{\text{number of rings } (N = 12) \times 12 \text{ tiles}}{\text{Total number of } C \text{ tiles}}$$

In this equation, the *number of rings* ($N=12$) was counted from AFM images. The *Total number of C tiles* was estimated from the sum of any sized rings and unassembled fragments. The detailed yield calculations were added in section 1 the methods of the supplementary information.

In the revised manuscript, we labeled the ring dimension on schematics only based on theoretical number calculated from B-form DNA models. Following the reviewer's advice, we reported bar charts to show the nanoring formation yields and distributions based on the monomer numbers in the measured nanorings. We also included a Supplementary Excel file containing detailed statistics for reference, such as the number of monomers counted for each sample.

3-3. The references regarding strand displacement on tile-based DNA structures need to be improved. A few that I think should be cited are:

Feng, Liping, et al. "A two-state DNA lattice switched by DNA nanoactuator." *Angewandte Chemie* 115.36 (2003): 4478-4482.

Zhang, David Yu, et al. "Integrating DNA strand-displacement circuitry with DNA tile self-assembly." *Nature communications* 4.1 (2013): 1965.

Green, Leopold N., et al. "Autonomous dynamic control of DNA nanostructure self-assembly." *Nature chemistry* 11.6 (2019): 510-520.

Response: Thanks for the suggestions. We cited these three papers as reference 23, 24, 25 in the revised manuscript, respectively.

3-4. I am confused by the choice of nomenclature throughout the paper. In Fig. 1a, middle, I do not understand why any of the monomers on the top row can be called DX, which stands for double crossover. I only see one crossover in most of them. For the 2 and 3 layer tiles, it is not clear what are the interactions of the helices in parallel laying on the plane perpendicular to the page. Are there any crossovers there? In Fig. 1b, left, this tile has 3 crossovers in the vertical arm, and only one strand crosses over at each point. Again, I am confused why this is called a DX tile. I am also very confused by what are 2-layer vs 3-layer tiles. The 1 layer is its own class, since in that case all helices appear stacked on the xy plane. But when we go to the 2-3 layer, what is the

number (2 or 3) referring to, and with respect to what are we layering? Please add a paragraph where this is clarified.

Response: Thank you for pointing it out. We agree that 'DX-T' is not accurate to name these tiles. In the revised manuscript, we use 'TC tile', which is an abbreviation of T-shape crossover tile. For the two- and three-layer tiles, we added schematics to display the side view and 3D view in Figure 1 and 2 in the revised manuscript. A simple example is shown in Figure R9.

Figure R9. Schematics of layers in TC tiles. The dash line indicates the layer. The center of DNA helix is used to assign layers.

For the two- and three-layer TC tiles, there are crossovers between helices in parallel laying on the plane. As shown in Figure R10 (Supplementary Fig 21&25 in revised manuscript), the line models clearly display all the crossovers in TC tiles.

Figure R10. Structural design of two- (a) and three-layer (b and c) DX-T tiles showing crossovers.

3-5. In Fig. 2, I am confused by the order of presentation of the tiles. In my opinion, it would make sense to present them in order of complexity, with Tiles 3, 5, and 6 going first, and later Tiles 1 (which includes 2 monomers of Tile 3 arranged to be symmetric relative to the y axis) and Tile 2 (which includes 2 connected Tile 4). The authors themselves acknowledge that the structures formed by Tiles 1 and 2 are identical to those achieved by Tiles 3 and 4 (which appear to form larger lattices). It would help to clarify why two approaches were pursued to assemble the same structure. I also note that the authors consider the presence of tubular assemblies (for Tiles 1 and 5 in particular) to be a nuisance, whereas I think it's a feature of their design that would be interesting to explore.

Response: Thanks for the suggestion. Before reorganizing the order of presentation of the tiles, we updated all the tiles with both linear and 2D self-assembly. The previous Figure 2 and 3 were combined as one figure. As suggested by the reviewer, we arranged all the tiles in order of complexity in Figure 2. The tiles that assemble into ladders were named 'Tile-X-1', while the corresponding tiles for 2D array were named 'Tile-X-2'. In the revised manuscript, we rearranged the figures by moving original Tile 1 and 2 to supplementary information and combining the previous Figures 2 and 3 into one figure. We also improved the logical flow to discuss these tile

designs in the revised manuscript (please refer to the 'Results and discussion' section of the manuscript).

We also explored the tubular formation for Tile 1 (TC-8-3 in the current manuscript) and Tile 5 (TC-3-2 in the current version) by changing the sticky end length. The data was summarized in the response to the question of 2-2 of reviewer 2 (Fig R4&5). The corresponding discussion is also updated in the revised manuscript (Supplementary Fig 19&20).

3-6. In Fig. 3 - Here I realized that the color coding of domains in the second row of schematics is inconsistent, then I looked back at Fig. 1 and realized they are inconsistent even there. E.g. what is called domain 1 is in dark blue in Tiles 7, 8, 11, then light blue in Tiles 10 and 12. Colors are inconsistent also for the other domains 2 and 3. Perhaps this doesn't matter, but it is confusing for the readers. Tile 7 appears identical to Tile 5 in Fig. 2. The authors say there is a different sticky end matching rule, but I could not figure out what it is based on the domain color/nomenclature used here. Finally, why is Tile 7 considered 2-layer but Tile 5 is not? This goes back to the nomenclature confusion I mentioned at point 4.

Response: We appreciate the reviewer's detailed comments. We changed the previous color-coding domains in the schematics into numbers or numbers with underlines to distinguish the sticky end matching rules (Fig 2). The panel b of schematics in Supplementary Fig 1-30 also shows the sticky end matching rules.

It was a typo to label Tile 7 as a one-layer tile in the previous manuscript. We corrected this mistake in the new revision.

3-7. Fig. 5 and 6: Which DNA strands were added to achieve each state transition? For Fig. 4, we have Fig. S19 to clarify which strands were added. Something similar is needed for Figs 5 and 6. Strands for Fig. 5 and 6 do not seem to be reported in any SI table either. If the experiments in Fig. 5 and 6 were done recombining strands used for previous figures, it needs to be made clear how.

Response: For Figures 5 and 6, we added a Supplementary Fig 43 to illustrate the details between transition states. The four groups of dynamic rings shared the same invaders strands and setting strands. We updated the strand sequences (Section 3 Supplementary notes). The Supplementary Excel file provides information on which step adds which strands.

The criterion for adding a number to each arrow of the pathway in Figs 5 and 6 does not seem to follow any logic nor match with numbering in Fig. 4, and the names of the SDR strands reported in SI table 4-5-6 (which refer to Fig. 4 anyway) does not closely follow the numbering in the main paper.

Response: We aligned the numbering rules in both main text and SI. We also updated the names of the SDR stands by following the numbering rule in the revised main paper.

For the state changes described in Fig 6, the authors note that there is a change in the number of monomers present in a ring: "Due to the change of monomer number in each ring, individual Ring 1 split into three of Ring 2 (or Ring 3) or five of Ring 4, showing an intriguing self-division process." I cannot agree with this statement given the poor description available. I am not even sure the sentence makes sense - do the authors want to say that Ring 1 splits into 3 or 4 other rings because of the change of monomers in a ring? The reason for diameter change should be

the addition of invader/settling strands, that then causes the change in monomer number. Anyway: there is no assessment of whether the rings are splitting, or rather they are disassembling into monomers, which are then conformationally switched and allowed to reassemble. So, I think it is misleading for readers to claim that the rings are splitting.

Response: We highly appreciate the reviewer's suggestions and comments. We have studied a series of reconfiguration processes by imaging intermediate structures during strand displacement reactions and summarized the new data into the revised manuscript (Fig 4, and Supplementary Fig. 35, 47&48). In total, 26 intermediate structures were observed. In the answer to question 1-3 of reviewer 1, we summarized 3 sets of reconfiguration processes (Fig R1-3).

Particularly, we highlighted the transformation between large rings and small rings here (Fig R2). We studied the 8-monomer rings resized into 5-monomer rings. After adding invader strands to trigger reconfiguration, we directly imaged a series of samples at different time points including 0, 1, 5, 30, and 120 minutes. As shown in Figure R2, most of the rings started to break into curved ladders after 1 minute. Smaller rings (N=4 or 5) can be observed even at 1 minute. We observed an intermediate structure, where a curved ladder adopted curvatures half from the large ring (N=8) and half from the small ring (N=5) (left zoom-in image in 5 min). In 120 minutes, most of the 8-monomer rings disappeared and smaller rings were obtained. After adding setting strands, the 8-monomer rings were displaced by small rings and byproducts structures.

Based on this experimental data, the original rings break open to adopt a different curvature for the target new rings, rather than disassembling into monomers. We have incorporated these new data into the revised manuscript and updated the corresponding description.

Minor:

3-8 There is often a mention of "vibration" of arms, but it is not clear in which direction/plane this is happening, and I didn't find this explanation helpful.

Response: Thank you for pointing this out. "Vibration of arms" was used three times in the previous version of the manuscript. We revised them one by one in the revised manuscript.

Previous: 'The flattened grids suggested that the introduction of strut structure helped to fix the in-plane flexible angle of Tile 2, while the unwrapped tubular arrays formation may be due to the off-plane vibration of vertical arms.'

Revised: 'The micrometer-sized 2D arrays yielded from TC-2-2 were generally larger than those from TC-1-2, suggesting that the struts helped to define the angle between arms and also corrected possible curvature of the structures.'

Previous: 'The two vertical helices of Tile 6 bind to the same helix of horizontal DX arm through three joint spots that can eliminate the in-plane vibration of the vertical arm (Fig. S 16a), while the two vertical helices of Tile 11 bind to two horizontal helices separately with two joint spots that cannot inhibit the in-plane vibration of vertical arm (Fig. S 16b).'

Revised: 'The vertical arms of TC-4-2 were connected to the same horizontal arm through one joint on T-loop and one joint on a DNA helix (Supplementary Fig 24). In comparison, the vertical arms of TC-5-2 bound to the horizontal arms separately with two joints on T-loops only, leading to a more flexible structure (Supplementary Fig 24).'

3-9 The authors need to explain how the yield was calculated and over how many experiments in the main paper. I found an explanation of how yield was computed in the caption of SI Fig.

18. It's not clear whether the classification of well-formed or misformed ring was done by hand or through an image processing software.

Response: We appreciate this suggestion. For the yield calculation, we used four of 3 μm \times 3 μm sized AFM images. The counting and classification of well-formed or mis-formed structures were done by hand. The detailed calculation equation was listed in section 1 the methods of supplementary information.

3-10 The figures in the main paper show up ok in the pdf but were small and very pixelated when printed. In general, I found the tile diagrams in the SI much easier to understand as one can follow the backbone of each individual strand marked in a different color. A consistently annoying thing in the SI is that captions of a figure are often in the next page.

Response: Thanks for pointing this out. We have improved the resolution of the main figures. The figures and captions in SI were rearranged to make them in the same page.

3-11 I found a few grammatical errors and expressions that can be improved. Please proofread and check the text.

Response: Thanks for the suggestion. We have corrected grammatical errors and proofread the text by all authors.

Reference

1. Dey, S., et al., *A reversibly gated protein-transporting membrane channel made of DNA*. Nat Commun, 2022. **13**(1): p. 2271.
2. Singh, J.K.D., et al., *Binding of DNA origami to lipids: maximizing yield and switching via strand displacement*. Nucleic Acids Res, 2021. **49**(19): p. 10835-10850.
3. Xing, Y., et al., *Functional Nanopores Enabled with DNA*. Angew Chem Int Ed Engl, 2023. **62**(33): p. e202303103.
4. Thomsen, R.P., et al., *A large size-selective DNA nanopore with sensing applications*. Nat Commun, 2019. **10**(1): p. 5655.

REVIEWER COMMENTS

Reviewer #1 (Remarks to the Author):

All the major concerns of this reviewer are addressed in the revision and a publication of the revised manuscript is recommended.

Reviewer #2 (Remarks to the Author):

The authors have addressed my concerns and suggestions. I recommend to publish this manuscript.

One minor comment is that I could not find figure captions for the main text.

Reviewer #3 (Remarks to the Author):

The authors revised the manuscript majorly and addressed all the comments I provided. The manuscript is much more readable and the motivation and impact have been clarified. I still suggest that the authors explain better what 2 layer vs 3 layer means in the TC tile design. I understand the designs are different, I just want to clarify why the nomenclature "layer" since I don't see what layer we are referring to. I asked this in the first round but it wasn't addressed.

I find it a bit strange that the authors added to the manuscript new nanoring designs based on single stranded tiles (SST), since the paper main message is the design of their TC tiles. I am not sure reviewer 1 was asking for a new SST designs, and this new figure seems a bit off-topic. How were these monomers designed and how do they relate to the TC tiles?

I also wish it was made clear what strands were added to achieve each numbered transition in Fig. 6. I think this information should be provided somewhere, but it was not obvious where to find this information in the SI.

The revised figures look a lot better but it seems they came without a caption, at least I was

not able to find it in the files shared with reviewers. I don't need to review the captions, but they have to be examined by the editors.

The authors have removed the DX notation as per my suggestion, and replaced it with AX. However the abstract still begins with the words "DNA double crossover tiles serve as the fundamental building blocks for DNA self-assembled nanostructures". This sentence should be changed as it is not true (there are dozens of other fundamental building blocks around), and puts in the spotlight the double crossover aspect that that is not really central to the paper.

REVIEWER COMMENTS

Reviewer #3 (Remarks to the Author):

1. The authors revised the manuscript majorly and addressed all the comments I provided. The manuscript is much more readable and the motivation and impact have been clarified. I still suggest that the authors explain better what 2 layer vs 3 layer means in the TC tile design. I understand the designs are different, I just want to clarify why the nomenclature "layer" since I don't see what layer we are referring to. I asked this in the first round, but it wasn't addressed.

Response: We thank the reviewer's comments and agree that it is essential to provide a clearer explanation of the distinction between 2-layer and 3-layer TC tile designs. As shown in **Figure R1**, we have labeled the layers as numbers 1, 2, and 3, with each dashed line denoting the center of an individual layer. To establish a coordinate reference, we have oriented the horizontal arm of the TC tile along the X-axis and the vertical arm along the Y axis, corresponding to the paper plane's XY axis. The layer of the tile is defined based on the number of DNA helices along the Z-axis (the depth). In the case of a two-layer TC tile, both horizontal and vertical arms consist of parallel double helices, and these two helices are aligned along the Z axis, collectively accounting for two layers. In contrast, the three-layer TC tiles have horizontal arms composed of parallel double helices, while the vertical arm is located spatially between the two horizontal helices, not belonging to either of these two layers, resulting in a total of three layers.

We have added new schematics to provide the side views and 3D representations in Figure 1 and 2 in the revised manuscript, along with Supplementary Figures 21 and 25.

Figure R1. Schematics of the layers in TC tiles. The layers are labeled as numbers 1, 2, and 3, with each dashed line denoting the center of an individual layer.

2. I find it a bit strange that the authors added to the manuscript new nanoring designs based on single stranded tiles (SST), since the paper main message is the design of their TC tiles. I am not sure reviewer 1 was asking for a new SST designs, and this new figure seems a bit off-topic. How were these monomers designed and how do they relate to the TC tiles?

Response: Thanks for the suggestion. The single-stranded tiles (SST) nanoring designs are based on C-shape TC tiles, in which T-junction and crossover are preserved. We added a supplementary Figure 36 (Figure R2) to show the design details of the 6-monomer and 18-monomer SST nanoring. To better clarify this, we revised the corresponding main text (second sentence of the fifth paragraph in the ‘2-state DNA nanoring reconfiguration’ section):

Previous: “We designed and demonstrated the formation and reconfiguration of a 6-monomer nanoring and an 18-monomer nanoring (Fig 5).”

Revised: “Based on C-shape TC tiles, we created 6-monomer and 18-monomer nanorings, and further demonstrated the formation and reconfiguration of them (Fig 5 & Supplementary Fig 36).”

The SST study is a natural extension of the non-SST tiles. In the non-SST designs, the assembled nanoring size was determined by the inherent geometry of the repeating tile units, which share the same geometry and sequence. In the SST designs, the SST nanorings were designed with well-defined sizes and addressable surfaces because the nanoring size is determined by both the geometry of each tile and the specific sequence design of each tile. For example, six tiles with distinct sequences assembled into a 6-monomer nanoring. Omitting one monomer or adding one more would prevent the creation of a nanoring due to the matching rule of the sticky end sequences. Therefore, SST tiles based on a C-shape TC design allowed us to have finer control over nanoring dimensions.

Figure R2. Schematics of the reconfigurable SST nanorings. a) The C-shape TC tiles that were used for SST nanoring. The tile for state 1 is TC-C-6-2.5-3 and state 2 is TC-C-4-2.5-3. b) Two-state SST nanorings. The 6-monomer nanoring consisted of six TC-C-6-2.5-3 monomers with different color-coded sequences, with well-defined matching rules between sticky ends.

3. I also wish it was made clear what strands were added to achieve each numbered transition in Fig. 6. I think this information should be provided somewhere, but it was not obvious where to find this information in the SI.

Response: We appreciate the reviewer’s suggestions. In the previous version, we summarized the information of the strands that were added to achieve each numbered transition in the supporting Excel file and Supplementary Fig 34. The sequences of invader strands and setting strands can be found in section 3 of SI. As the reviewer suggested, we have added Supplementary Table 13 and Table 14 (Table R1 & R2) in the revised Supplementary Information (SI) to illustrate the strand information for each transition of single monomer nanoring and SST nanoring.

Table R1. The invader and setting strands of each transition between four pairs of nanorings.

pair	path	State 1	State 2	invader strand	setting strand
		monomer tile	monomer tile		
pair 1	path 1	TC-C-7-3.5-6	TC-C-7-3.5-4	invader strand 2	setting strand 2
	path 2	TC-C-7-3.5-6	TC-C-9-3.5-6	invader strand 3	setting strand 3
	path 3	TC-C-7-3.5-6	TC-C-9-3.5-4	invader strand 2&3	setting strand 2&3
	path 4	TC-C-9-3.5-6	TC-C-7-3.5-6	invader strand 1	setting strand 1
	path 5	TC-C-9-3.5-6	TC-C-9-3.5-4	invader strand 2	setting strand 2
	path 6	TC-C-9-3.5-6	TC-C-7-3.5-4	invader strand 1&2	setting strand 1&2
	path 7	TC-C-7-3.5-4	TC-C-9-3.5-6	invader strand 3&4	setting strand 3&4
	path 8	TC-C-7-3.5-4	TC-C-7-3.5-6	invader strand 4	setting strand 4
	path 9	TC-C-7-3.5-4	TC-C-9-3.5-4	invader strand 3	setting strand 3
	path 10	TC-C-9-3.5-4	TC-C-7-3.5-4	invader strand 1	setting strand 1
	path 11	TC-C-9-3.5-4	TC-C-9-3.5-6	invader strand 4	setting strand 4
	path 12	TC-C-9-3.5-4	TC-C-7-3.5-6	invader strand 1&4	setting strand 1&4
pair 2	path 1	TC-C-7-5.5-6	TC-C-7-5.5-4	invader strand 2	setting strand 2
	path 2	TC-C-7-5.5-6	TC-C-9-5.5-6	invader strand 3	setting strand 3
	path 3	TC-C-7-5.5-6	TC-C-9-5.5-4	invader strand 2&3	setting strand 2&3
	path 4	TC-C-9-5.5-6	TC-C-7-5.5-6	invader strand 1	setting strand 1
	path 5	TC-C-9-5.5-6	TC-C-9-5.5-4	invader strand 2	setting strand 2
	path 6	TC-C-9-5.5-6	TC-C-7-5.5-4	invader strand 1&2	setting strand 1&2
	path 7	TC-C-7-5.5-4	TC-C-9-5.5-6	invader strand 3&4	setting strand 3&4
	path 8	TC-C-7-5.5-4	TC-C-7-5.5-6	invader strand 4	setting strand 4
	path 9	TC-C-7-5.5-4	TC-C-9-5.5-4	invader strand 3	setting strand 3
	path 10	TC-C-9-5.5-4	TC-C-7-5.5-4	invader strand 1	setting strand 1
	path 11	TC-C-9-5.5-4	TC-C-9-5.5-6	invader strand 4	setting strand 4
	path 12	TC-C-9-5.5-4	TC-C-7-5.5-6	invader strand 1&4	setting strand 1&4
pair 3	path 1	TC-Z-7-4-6	TC-Z-7--4-4	invader strand 2	setting strand 2
	path 2	TC-Z-7-4-6	TC-Z-9--4-6	invader strand 3	setting strand 3
	path 3	TC-Z-7-4-6	TC-Z-9--4-4	invader strand 2&3	setting strand 2&3
	path 4	TC-Z-9-4-6	TC-Z-7-4-6	invader strand 1	setting strand 1
	path 5	TC-Z-9-4-6	TC-Z-9--4-4	invader strand 2	setting strand 2
	path 6	TC-Z-9-4-6	TC-Z-7--4-4	invader strand 1&2	setting strand 1&2
	path 7	TC-Z-7-4-4	TC-Z-9--4-6	invader strand 3&4	setting strand 3&4
	path 8	TC-Z-7-4-4	TC-Z-7--4-6	invader strand 4	setting strand 4
	path 9	TC-Z-7-4-4	TC-Z-9--4-4	invader strand 3	setting strand 3
	path 10	TC-Z-9-4-4	TC-Z-7--4-4	invader strand 1	setting strand 1
	path 11	TC-Z-9-4-4	TC-Z-9--4-6	invader strand 4	setting strand 4
	path 12	TC-Z-9-4-4	TC-Z-7--4-6	invader strand 1&4	setting strand 1&4
pair 4	path 1	TC-Z-8-4-6	TC-Z-6--4-4	invader strand 2	setting strand 2
	path 2	TC-Z-8-4-6	TC-Z-8--4-6	invader strand 3	setting strand 3
	path 3	TC-Z-8-4-6	TC-Z-8--4-4	invader strand 2&3	setting strand 2&3
	path 4	TC-Z-8-4-6	TC-Z-6--4-6	invader strand 1	setting strand 1
	path 5	TC-Z-8-4-6	TC-Z-8--4-4	invader strand 2	setting strand 2
	path 6	TC-Z-8-4-6	TC-Z-6--4-4	invader strand 1&2	setting strand 1&2
	path 7	TC-Z-8-4-4	TC-Z-8--4-6	invader strand 3&4	setting strand 3&4
	path 8	TC-Z-8-4-4	TC-Z-6--4-6	invader strand 4	setting strand 4
	path 9	TC-Z-8-4-4	TC-Z-8--4-4	invader strand 3	setting strand 3
	path 10	TC-Z-8-4-4	TC-Z-6--4-4	invader strand 1	setting strand 1
	path 11	TC-Z-8-4-4	TC-Z-8--4-6	invader strand 4	setting strand 4
	path 12	TC-Z-8-4-4	TC-Z-6--4-6	invader strand 1&4	setting strand 1&4

Invader and setting strands for 4-state nanoring transformation.

Invader strand 1: GCTATTGGTGATGTCCTGTTTTTACGGACATTTCAACGTTAAATCCAAGG
Invader strand 2: ACAGACACAGAGACACTTTTTTGGTCTCAACAAGCATCATCTCTAGGTT
Invader strand 3: GCTATTGGTGATCAACGTTAAAGTGATGG
Invader strand 4: ACAGACACACAAGCATCATCTACAGTTG
setting strand 1: CCATCACTTTAACGTTGATCACCAATAGC
setting strand 2: CAACTGTAGATGATGCTTGTG TGTCTGT
setting strand 3: CCTTGGATTTAACGTTGAAATGTCCGTAAAAACAGGACATCACCAATAGC
setting strand 4: AACCTAGAGATGATGCTTGTGAGACCAAAAAAGTGTCTCTGTGTCTGT

Table R2. The invader and setting strands of SST nanoring transformation.

	State1 monomer tile	State 2 monomer tile	invader strand	setting strand
SST	TC-C-6-2.5-3	TC-C-4-2.5-3	invader strand 1*	setting strand 1*
nanoring	TC-C-4-2.5-3	TC-C-6-2.5-3	invader strand 2*	setting strand 2*

Invader and setting strands for SST ring transformation.

Invader strand 1*: ACAGACACAGAGACACTTTTTTGGTCTCAACAAGCATCATCTCTAGGTT
setting strand 1*: CAACTCTAGATGATGCTTGTGTGTCTGT
Invader strand 2*: ACAGACACACAAGCATCATCTAGAGTTG
setting strand 2*: AACCTAGAGATGATGCTTGTGAGACCAAAAAAGTGTCTCTGTGTCTGT

4. The revised figures look a lot better but it seems they came without a caption, at least I was not able to find it in the files shared with reviewers. I don't need to review the captions, but they have to be examined by the editors.

Response: Thank you for pointing this out. We have uploaded the figures with captions separately in the submitting system. To facilitate the reviewing process, we have attached the figures with captions at the end of the manuscript (after references) in the revised version.

5. The authors have removed the DX notation as per my suggestion, and replaced it with AX. However the abstract still begins with the words "DNA double crossover tiles serve as the fundamental building blocks for DNA self-assembled nanostructures". This sentence should be changed as it is not true (there are dozens of other fundamental building blocks around), and puts in the spotlight the double crossover aspect that that is not really central to the paper.

Response: We appreciate the reviewer's viewpoint and agree that the first few sentences of the abstract are not appropriate. We have revised the corresponding sentences.

Previous: "DNA double crossover tiles serve as the fundamental building blocks for DNA self-assembled nanostructures. Leveraging the design principle of bundling DNA helices through crossovers, these tiles have found extensive application in various nano-constructions such as DNA arrays, designer crystals, and DNA origami. Introducing additional binding arms with programmable angles to the parallel-packed helices of these DNA tiles will enable new nano-assemblies and potential uses."

Revised: "DNA tiles serve as the fundamental building blocks for various DNA self-assembled nanostructures such as DNA arrays, designer crystals, and DNA origami. Creating new DNA tiles by introducing additional binding arms with programmable angles to DNA crossover tiles holds the promise of unlocking novel nano-assemblies and potential applications."

REVIEWERS' COMMENTS

Reviewer #3 (Remarks to the Author):

The authors have addressed my comments and I don't have further suggestions for improvement.